# SEIQRS model analysis and optimal control with two delays

**Junling Wang◉, Lei Zhong◉‡\*, Xinxin Chang‡**

School of Information Engineering, Jiangxi University of Science and Technology, Ganzhou, Jiangxi, China

◉ These authors contributed equally to this work.
‡ LZ and XC also contributed equally to this work.
\* 6120210176@mail.jxust.edu.cn

## Abstract

This paper addresses the limitations of assuming a bilinear infection rate in computer virus propagation models by proposing a more realistic nonlinear dual-delay SEIQRS model. The key focus of the research includes analyzing the local stability of the disease-free equilibrium point and the existence of positive equilibrium points with different delays. To further enhance control effectiveness, time-varying control terms are introduced, and a corresponding Hamiltonian function is constructed. The optimal control strategy is derived using the Pontryagin's maximum principle. Numerical simulation experiments are conducted to validate the dynamic characteristics of the model, and the model's properties are verified on real-time propagation networks. The experimental results demonstrate that appropriate control strategies can effectively suppress the spread of computer viruses.

## 1. Introduction

In the context of the current information age, the rapid advancements in modern science and technology, particularly in the realm of computers and electronic communications, have brought about significant transformations in both production processes and daily lifestyles. The Internet, in particular, has emerged as an indispensable and integral component of contemporary life. According to the 52nd Statistical Report on Internet Development in China [1] published by the China Internet Network Information Center (CNNIC), as of June 2023, the number of Internet users in China had reached 1.079 billion, reflecting an increase of 11.09 million since December 2022. Moreover, the Internet penetration rate had reached 76.4% in the country. The advent of the internet has brought about numerous conveniences for people, but it has also created favorable conditions for the rapid spread of computer viruses. Unlike biological viruses, computer viruses are artificially created program codes that search for other programs or media that meet their infection criteria when executed on a computer. Once a target is identified, the virus inserts its own code and proceeds to

**Data availability statement:** The experimental data of this paper are from this website: https://github.com/Zhonglei0029/SEIQRS.

**Funding:** This work is supported by Jiangxi Provincial Key Laboratory of Multidimensional Intelligent Perception and Control (CN).

**Competing interests:** The authors have declared that no competing interests exist.

replicate itself. If an infected file is not promptly addressed, it can have a significant impact on human life. Therefore, researching the propagation patterns and control strategies of computer viruses holds great significance and profound implications. It is in this context that the field of computer virus epidemiology has emerged.

Computer virus epidemiology is an interdisciplinary field that investigates the propagation patterns and control strategies of computer viruses in networks. It draws upon methods and models from disciplines such as epidemiology, complex network theory, and dynamical systems. This field provides a theoretical foundation and technical means for preventing and mitigating network security threats. Scholars both domestically and internationally primarily focus on propagation models and mechanisms. They employ mathematical models and complex network theory to describe the spreading process of viruses in networks and explore the patterns and mechanisms of virus propagation. This research enables the development of corresponding defense strategies, which are of crucial significance in formulating virus defense strategies and enhancing network security.

(1) Research on computer virus propagation models

Research on the dynamics of computer virus propagation can be traced back to the late 1980s. In the late 1980s, F. Cohen [2] published a paper elucidating the existence of computer viruses and proposed the possibility of studying the propagation patterns of computer viruses using dynamic models borrowed from the field of infectious diseases in medicine. This work garnered significant attention from the academic community. In 1991, J.O. Kephart and S.R. White [3], drawing inspiration from a class of classical infectious disease models known as the SIS compartmental model, successfully established the first compartmental model for computer virus propagation. Subsequently, with the widespread adoption of the Internet, computer virus propagation entered a new stage. Researchers began collecting empirical data on actual virus propagation and conducting large-scale empirical analyses to understand the characteristics and patterns of virus propagation. During this stage, propagation models evolved beyond theoretical frameworks and became more practical, taking into account factors such as network topology and user behavior.

With the in-depth study of complex network theory, researchers have started applying it to computer virus propagation models. Complex network models can better describe the structure and characteristics of real networks, thereby improving the accuracy of virus propagation models. In 2020, Chenquan Gan et al. [4] studied the propagation behavior of computer viruses on complex networks under the joint influence of network topology structure and mobile storage media. They conducted a carefully designed set of experiments on the Oregon routing network and found that mobile storage media have a significant impact on virus propagation. Ronghua Shang et al. [5] proposed a dynamic node immunization model based on community structure and thresholds (NICT). This model considers nodes carrying harmful information as new nodes in the network and changes the way new nodes establish edges with original nodes based on the needs of different networks. They determine the propagation probability between nodes by using community structure information

and a similarity function between nodes, and propose an improved immunization gain. Experimental results show that this method can provide better network immunity. In 2022, Zhang Ningbo et al. [6] proposed a hybrid resource allocation scheme and established a virus-resource asymmetric coupling propagation model to simulate the virus propagation process under the influence of resource recovery. They used a generalized discrete Markov chain approach to describe the dynamics of propagation. Juan Liu [7] studied the propagation patterns of malicious code in wireless sensor networks and established a prevalence model for wireless sensor networks with graded infection rates and two time delays. They discussed the impact of the two time delays on propagation, providing reference and effective suggestions for network administrators. In 2023, Rahat Zarin et al. [8] combined fractional calculus with the nonlinear interactions between nodes in wireless sensor networks and proposed a nonlinear fractal fractional mathematical model for analyzing the propagation of malicious code in wireless sensor networks. They elucidated the propagation patterns of viruses in vulnerable systems and potential countermeasures. In 2024, Kun Li et al. [9] combined evolutionary game theory with complex network theory to study the competition effect of different travel strategies on virus propagation. Simulation results show that self-isolation strategies can significantly inhibit the spread of infections in complex social networks, and the introduction of reward mechanisms further enhances this effect. Additionally, a higher network degree favors the prevalence of self-isolation, thereby impeding virus propagation.

In recent years, with the development of artificial intelligence and data science, researchers have started incorporating behavior modeling into computer virus propagation models. This type of model takes into account factors such as user behavior, social network relationships, etc., to more accurately predict the path and speed of virus propagation. In 2020, Dang Q.A. et al. [10] proposed a new method based on the Lyapunov stability theorem and its extension, combined with the global stability theorem of discrete-time nonlinear fractional-order systems, to theoretically prove that the proposed NSFD (Non-Standard Finite Difference) scheme preserves the global asymptotic stability of the original model. They also presented the positivity and global stability preservation of the NSFD scheme for computer virus mixed propagation models. In 2021, Shashank Awasthi et al. [11] considered the charging mechanism of low-energy sensor nodes and proposed the SILRD model, which describes the dynamic propagation of malicious software and the energy consumption of sensor nodes due to malicious software attacks. They also proposed defense techniques for preventing the propagation of malicious software in wireless sensor networks. In 2022, Nguyen Phuong Dong et al. [12] studied the uncertainty attack behavior of computer viruses in wireless sensor networks using fuzzy fractional derivatives with non-local Mittag-Leffler function kernels. They proposed a fuzzy fractional SIQR model to describe the dynamics of virus propagation in the network and discussed the concept of fuzzy fractional derivatives in the Caputo sense. They also discussed important properties of the fuzzy fractional SIQR model with fuzzy data. In 2023, Guowen Wu [13] proposed a new virus propagation model called STSIR based on the analysis framework of epidemiology and individual-group game theory. This model considers human behavior and introduces an individual-group game to establish an attack and defense model between infected SIoT (Social Internet of Things) nodes and susceptible SIoT nodes. By using the payoff matrix to obtain mixed Nash equilibrium solutions, the model provides a more reasonable description of virus propagation among devices.

(2) Research on computer virus spread control

In the context of biological virus models in the real world, the essential factors for infectious diseases are the infectious source, transmission routes, and susceptible individuals. In the biological realm, susceptible individuals refer to the organisms that can be infected. Similarly, for models of computer virus propagation, the elements of computer virus propagation in a network are infectious computer hosts, transmission routes, and susceptible computers. Therefore, most computer virus control models are studied from these aspects.

In 2015, Ahn et al. [14] proposed the C-SEIRA model for the propagation of malicious worms. They applied optimal control theory to minimize the number of infected hosts and the system processing cost of isolating infected computers from the network as objectives. This approach effectively controlled the spread of malicious worms. In 2018, Abu Arqub

et al. [15] proposed a novel approach to simplify many problems encountered in various scientific and engineering fields by discretizing integral-differential algebraic equation systems or their corresponding system sequences. This approach provided a new perspective for establishing computer virus propagation models. In 2020, Upadhyay et al. [16] introduced two different frameworks, namely attack class and target class models, to study the mobility of viruses in target networks. They proposed a new model for controlling DDoS attacks and utilized optimal control theory to regulate virus propagation and reduce infection levels. In 2021, Xingguo Li et al. [17] presented an intermittent isolation-immune strategy for controlling virus propagation in the presence of information diffusion. In this model, information and viruses propagate and interact in different subnetworks. They developed a heterogeneous mean-field approach with time delay to study the dynamics of virus spread, and systematically investigated the propagation dynamics using Monte Carlo simulations. The results demonstrate that the optimal information transmission probability of the virus will be significantly suppressed during relatively short intermittent periods. Conversely, increasing the probability of information transmission can control virus propagation and inhibit the increase of intermittent periods when the periods are very long. Abu Arqub et al. [18] employed the reproducing kernel Hilbert space method with error estimation and convergence analysis to solve the Fredholm-constrained optimal control problem, providing strong theoretical support for controlling the spread of computer viruses. In 2022, Shahini et al. [19] proposed a method for solving the optimal control problem (OCP) of nonlinear time-delay differential equations at Legendre-Gauss-Radau points. They utilized operational matrices and collocation method, combined with a hybrid of Legendre polynomials and block pulse functions to discretize the model, transforming it into a large-scale finite-dimensional nonlinear programming problem. Experimental results indicate that strategies that increase failure rates and retrieval rates while reducing prevalence rates and the number of new computers have a significant impact on controlling and suppressing the spread of computer viruses. Hind Sweis et al. [20] transformed the fractional delay problem into a Volterra integral equation. They established a local existence theorem using the Arzelà-Ascoli theorem and Schauder fixed-point theorem and investigated ρ-order fractional delay differential equations with Atangana-Baleanu fractional derivatives in the Caputo sense. Their research provided strong support for stability studies of computer virus models. In 2023, Ruiling Wang et al. [21] introduced the exposed partitions of computers and USB devices based on the potential characteristics of worms. They proposed a series of computer worm propagation models with saturation occurrence rates and studied the qualitative behavior of the proposed models. Furthermore, they extended the worm propagation models to include three control strategies aimed at minimizing the number of computers and USB devices carrying computer worms. Qingyi Zhu et al. [22] presented a new mathematical model for the propagation of industrial viruses in SCADA systems and established an optimal control system for the model. Through optimal control, they achieved better control of virus propagation with limited resources. Haesung Ahn et al. [23] addressed the significant challenges posed by popular malicious software such as Stuxnet to autonomous vehicles. They developed a mathematical model for the temporal and spatial propagation of Stuxnet-style malware and discussed the future applications of the model in the temporal and geographical spread of infectious malware based on AVs, providing new insights for future research. Qianhui Zhuang et al. [24] proposed a malicious virus propagation model with diffusion and time delay for Cyber-Physical Systems (CPS) and introduced a proportional derivative (PD) feedback control scheme. Experimental results demonstrated that the PD control scheme successfully suppressed the instability of Turing machines and achieved the goal of regulating Hopf bifurcation, providing guidance for predicting and controlling the spread of malicious software. Avcı Derya [25] considered the possibility of failure in recycled computers due to mechanical reasons. They established a fractional-order SEIR computer virus propagation model and obtained numerical solutions for the optimal system using a fractional Euler method combined with forward-backward sweep algorithm. Their approach aimed to minimize the cost of antivirus software installation and eliminate the harm caused by virus propagation in computer networks. Pushpendra et al. [26] derived a mathematical model for the Flash Worm using an improved Caputo-type fractional-order derivative. They analyzed the stability of the model under different values of new computer arrival rate, damage rate, virus transmission rate, and natural clearance rate. Optimal control theory was employed to control the spread of the Flash Worm. Shen Shigen et al. [27] addressed the

characteristics of simple edge devices and limited central computer resources in Industrial Internet of Things (IIoT). They proposed a two-layer malicious software propagation patch model called IIPV based on a hybrid patch allocation method. They solved the optimization problem with subjective effort parameters using optimization theory to obtain the optimal control strategy for devices against malicious software and patches. Additionally, they integrated deep reinforcement learning algorithms into the IIPV model and designed a new algorithm called DDQN-PV, which effectively suppressed the propagation of malicious software in IIoT during experimental processes, providing a reference for future research.

Currently, many scholars have conducted extensive research on the dynamic models of computer virus propagation based on the mechanisms of computer virus transmission. However, the majority of these studies assume a bilinear infection rate. In reality, this assumption is not accurate because the increase in susceptible computers and infected computers makes it impossible to achieve a bilinear infection rate. Nonlinear infection rates are more realistic. In light of this, this paper constructs an SEIQRS computer virus propagation model with preventive security controls and conducts research on the following aspects of virus transmission:

1. Considering the influence of virus latency period and temporary immunity period on transmission: This paper investigates the impact of virus latency period and temporary immunity period on the speed and scale of virus transmission. By introducing these factors, the model more accurately reflects the reality of virus propagation.

2. Studying the stability of equilibrium points and the Hopf bifurcation phenomenon: This paper analyzes the local stability of the disease-free equilibrium point and the positive equilibrium point with delays in the model, and discusses the possible occurrence of Hopf bifurcation. These analyses contribute to understanding the dynamic characteristics of virus transmission.

3. Proposing optimal control strategies: This paper formulates an optimal control problem for the model, aiming to determine the optimal security control strategies to effectively mitigate the speed and scale of virus transmission. By applying mathematical methods, the existence of optimal control is determined, and practical and effective transmission control measures are provided.

4. Numerical simulation validation: To validate the effectiveness and accuracy of the model, numerical simulation experiments are conducted in this paper. By simulating virus transmission processes under different scenarios, the dynamic characteristics of the model and the effectiveness of control strategies are verified.

Through research in these areas, this paper aims to gain a better understanding of the propagation patterns of computer viruses and provide scientific basis for the development of prevention and control strategies. This paper is orgnized as follows. In Section 2, a two-delay SEIQRS with nonlinear incidence rates is used to represent the propagation of computer viruses. In Section 3, the local stability of the model's disease-free equilibrium point and the positive equilibrium point with various time delays are examined, and its Hopf bifurcation is discussed. The optimal control problem is presented in Section 4 by adding control criteria, and a workable and efficient control strategy for computer virus propagation is created. By using numerical simulations, Section 5 verifies the model's dynamic properties and demonstrates how, after enforcing an appropriate control approach, it is possible to effectively control the spread of computer viruses. The findings are summarized and discussed in Section 6.

## 2. Model establishment

V. MadhuSudanan et al. [28] developed a computer viral model with a nonlinear infection rate and infection cycle delay, taking into account that the network architecture in virus diffusion induces nonlinear infection.:

$$\begin{cases} \frac{dS(t)}{dt} = (1-p)b - \frac{\beta S(t-\tau_1)I(t-\tau_1)}{1+\sigma S(t-\tau_1)} - dS(t) + \delta R(t), \\ \frac{dI(t)}{dt} = \frac{\beta S(t-\tau_1)I(t-\tau_1)}{1+\sigma S(t-\tau_1)} - (d+\alpha+\gamma)I(t), \\ \frac{dR(t)}{dt} = pb + \gamma I(t) - (d+\delta)R(t), \end{cases} \tag{1}$$

Fangfang Yang et al. [29] introduced the isolation strategy based on the work of V. MadhuSudanan and R. Geetha and established a double-delay SIQRS computer virus model:

$$\begin{cases} \frac{dS(t)}{dt} = (1-p)b - \frac{\beta S(t-\tau_1)I(t-\tau_1)}{1+\sigma S(t-\tau_1)} - dS(t) + \delta R(t-\tau_2), \\ \frac{dI(t)}{dt} = \frac{\beta S(t-\tau_1)I(t-\tau_1)}{1+\sigma S(t-\tau_1)} - (d+\alpha_1+\gamma+\varepsilon)I(t), \\ \frac{dQ(t)}{dt} = \varepsilon I(t) - (d+\alpha_2+\eta)Q(t), \\ \frac{dR(t)}{dt} = pb + \gamma I(t) + \eta Q(t) - dR(t) - \delta R(t-\tau_2), \end{cases} \qquad (2)$$

The incubation period refers to the time interval between the invasion of a pathogen into the human body and the onset of clinical symptoms. It is influenced by various factors, including the quantity of the pathogen, its virulence, reproductive capacity, and the body's resistance. Similarly, computer viruses also possess the significant characteristic of an incubation period. In real-life situations, computer viruses require the execution of specific programs to become activated, and if these programs are not executed, the computer virus can remain dormant for an extended period. Therefore, when studying the propagation patterns of computer viruses, it is necessary to consider the impact of the incubation period on virus transmission. Furthermore, similar to biological viruses, coming into contact with an infected individual does not necessarily result in immediate infection. It takes some time of interaction with the infected individual to enter the incubation period. Hence, there is a time delay effect from the susceptible state to the latent state that needs to be adequately considered. However, existing models overlook these issues and are not sufficiently aligned with the real-life situation.

This work was inspired by the research of V. MadhuSudanan [28] and Fangfang Yang [29] as well as the aforementioned model. According to the warehouse theory [30], in this paper, all state nodes are divided into five compartments: $S(t)$, $E(t)$, $I(t)$, $Q(t)$ and $R(t)$. Where, $S(t)$ is the susceptible node, indicating the number of healthy nodes that do not have immunity and is vulnerable to virus infection at time $t$. $E(t)$ is a latent node, which means that after the node is infected with the virus at time $t$, there is no immediate outbreak and the number of nodes that cannot infect. $I(t)$ is the infected node, representing the number of nodes infected by the virus and having a specific infection capacity at time $t$. $Q(t)$ is the isolation node and represents the number of nodes that disconnect from the network and enter the isolation state at time $t$. $R(t)$ is the immune node, representing the number of nodes not capable of infection but entering the immune state at time $t$.

The state transition diagram of the computer virus spreading in the network is shown in Fig 1:

The rules for node status transfer are as follows:

1) Susceptible node $S \rightarrow$ Latent node $E$: Some susceptible nodes enter the latent state to be activated due to virus infection, and temporarily have no attack capability;

2) Latent node $E \rightarrow$ Susceptible node $S$: Some latent nodes are still not activated after the end of the network operation and become susceptible nodes again.

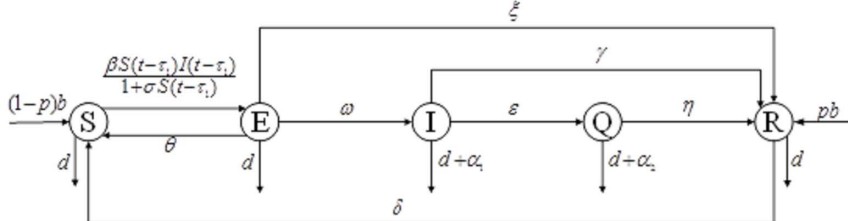

**Fig 1. SEIQRS model state transition diagram.**

3) Latent node $E\rightarrow$ Immune node $R$: Some latent nodes are transformed into immune state due to anti-virus software.

4) Latent node $E\rightarrow$ Infected node $I$: Viruses in some latent nodes are activated and become exposed nodes that can infect other network nodes.

5) Infected node $I\rightarrow$ Immune node $R$: Some infected nodes are transformed into immune nodes by anti-virus software.

6) Infected node $I\rightarrow$ Isolated node $Q$: The infected node in virus defence disconnects and enters the isolation state.

7) Isolated node $Q\rightarrow$ Immune node $R$: The node that enters the isolation state after being infected with the virus has the ability to antiviral attack infection after autonomic immune function, and enters the immune state;

8) Immune node $R\rightarrow$ Susceptible node $S$: Due to virus mutation and other reasons, the immune node cannot resist the new virus and is transformed into a susceptible node again.

The model description is shown in system (3), and the parameter description is shown in Table 1:

$$\begin{cases} \frac{dS(t)}{dt} = (1-p)b + \delta R(t-\tau_2) + \theta E(t) - \frac{\beta S(t-\tau_1)I(t-\tau_1)}{1+\sigma S(t-\tau_1)} - dS(t), \\ \frac{dE(t)}{dt} = \frac{\beta S(t-\tau_1)I(t-\tau_1)}{1+\sigma S(t-\tau_1)} - (\theta + d + \omega + \xi)E(t), \\ \frac{dI(t)}{dt} = \omega E(t) - (d + \alpha_1 + \gamma + \varepsilon)I(t), \\ \frac{dQ(t)}{dt} = \varepsilon I(t) - (d + \alpha_2 + \eta)Q(t), \\ \frac{dR(t)}{dt} = pb + \xi E(t) + \gamma I(t) + \eta Q(t) - dR(t) - \delta R(t-\tau_2), \end{cases} \tag{3}$$

The SEIQRS computer virus spreading model studied in this paper follows the dynamic equation of system (3) and assumes that the initial conditions that system (3) needs to satisfy are: $S(\theta) = \phi_1(\theta); E(\theta) = \phi_2(\theta); I(\theta) = \phi_3(\theta); Q(\theta) = \phi_4(\theta); R(\theta) = \phi_5(\theta);$ Among them:

$$\phi_i(\theta) \in ([-\tau, 0], R^5), \ \phi_i(0) > 0; \ R^5 = \left\{(x_1, x_2, x_3, x_4, x_5), \ x_i \geq 0\right\}, \ i = 1, 2, 3, 4, 5;$$

**Table 1. Model parameter description.**

| Parameter | Instructions |
|---|---|
| $b$ | Number of new nodes in the network. |
| $p$ | The immunization rate of new nodes. |
| $d$ | Regular offline rate of nodes in each state. |
| $\theta$ | The probability that the latent node is not activated. |
| $\beta$ | Node infection rate. |
| $\sigma$ | Measure the saturation coefficient of inhibition. |
| $\omega$ | The probability that a latent node is activated. |
| $\xi$ | The probability that the latent node is immune. |
| $\varepsilon$ | The probability that the exposed node disconnects from the network. |
| $\gamma$ | The probability that the antivirus software successfully detects the virus. |
| $\eta$ | The probability that the quarantined node successfully detects the virus and is transformed into an immune node. |
| $\delta$ | The immune loss rate of immune nodes. |
| $\alpha_1$ | Exposed node offline rate due to virus infection. |
| $\alpha_2$ | Rate of quarantined nodes going offline due to virus infection. |
| $\tau_1$ | Contact delay from susceptible state to latent state. |
| $\tau_2$ | The immune delay before an immune node becomes a susceptible node. |

Let $N(t) = S(t) + E(t) + I(t) + Q(t) + R(t)$. According to system (3) we get:

$$\frac{dN(t)}{dt} = \frac{dS(t)}{dt} + \frac{dE(t)}{dt} + \frac{dI(t)}{dt} + \frac{dQ(t)}{dt} + \frac{dR(t)}{dt}$$
$$= b - dS(t) - dE(t) - dI(t) - dQ(t) - dR(t) - \alpha_1 I(t) - \alpha_2 Q(t)$$
$$= b - dN(t) - \alpha_1 I(t) - \alpha_2 Q(t) < b - dN(t)$$

When $t \to \infty$, there is $0 \le N(t) \le \frac{b}{d}$. Let $\Omega = \left\{ (S, E, I, Q, R) \,|\, 0 < S + E + I + Q + R \le \frac{b}{d} \right\}$, $\Omega$ is the forward invariant set of system (3).

## 3. Study on stability of equilibrium point and Hopf bifurcation

Let $Z_1 = \theta + d + \omega + \xi$, $Z_2 = d + \alpha_1 + \varepsilon + \gamma$, $Z_3 = d + \alpha_2 + \eta$, $Z_4 = d + \delta$, then system (3) is simplified to system (4):

$$\begin{cases} \frac{dS(t)}{dt} = (1-p)b + \delta R(t-\tau_2) + \theta E(t) - \frac{\beta S(t-\tau_1) I(t-\tau_1)}{1+\sigma S(t-\tau_1)} - dS(t), \\ \frac{dE(t)}{dt} = \frac{\beta S(t-\tau_1) I(t-\tau_1)}{1+\sigma S(t-\tau_1)} - Z_1 E(t), \\ \frac{dI(t)}{dt} = \omega E(t) - Z_2 I(t), \\ \frac{dQ(t)}{dt} = \varepsilon I(t) - Z_3 Q(t), \\ \frac{dR(t)}{dt} = pb + \xi E(t) + \gamma I(t) + \eta Q(t) - dR(t) - \delta R(t-\tau_2), \end{cases} \tag{4}$$

Let $\frac{dS(t)}{dt} = 0$, $E(t) = 0$, $I(t) = 0$, $Q(t) = 0$, $\frac{dR(t)}{dt} = 0$, The disease-free equilibrium point of system (4) can be obtained: $P^0(S^0, E^0, I^0, Q^0, R^0) = \left( \frac{b(d-dp+\delta)}{dZ_4}, 0, 0, 0, \frac{pb}{Z_4} \right)$. Let $\frac{dS(t)}{dt} = 0$, $\frac{dE(t)}{dt} = 0$, $\frac{dI(t)}{dt} = 0$, $\frac{dQ(t)}{dt} = 0$, $\frac{dR(t)}{dt} = 0$, the positive equilibrium point of system (4) can be obtained:

$P^*(S^*, E^*, I^*, Q^*, R^*) = \left( \frac{Z_1 Z_2}{\omega \beta - \sigma Z_1 Z_2}, E^*, \frac{\omega}{Z_2} E^*, \frac{\omega \varepsilon}{Z_2 Z_3} E^*, \frac{pb}{d+\delta} + \frac{\xi}{d+\delta} E^* + \frac{\gamma \omega}{Z_2(d+\delta)} E^* + \frac{\eta \varepsilon \omega}{Z_2 Z_3(d+\delta)} E^* \right)$ with

$E^* = \left( -b + bp - \frac{\delta pb}{d+\delta} + \frac{dZ_1 Z_2}{\omega \beta - \sigma Z_1 Z_2} \right) \left( \frac{Z_2 Z_3(d+\delta)}{Z_2 Z_3 \delta \xi + Z_3 \delta \gamma \omega + \delta \eta \varepsilon \omega + Z_2 Z_3 \theta(d+\delta) - Z_1 Z_2 Z_3(d+\delta)} \right)$. In the propagation

dynamics, the primary reproduction number is the critical threshold to determine whether there is a virus in the system. According to the previous method [15], We can get the transfer-in matrix $F$ and transfer-out matrix $V$ of system (4) as follows:

$$F = \begin{pmatrix} 0 & \frac{\beta S(t)}{1+\sigma S(t)} \\ 0 & 0 \end{pmatrix}, \quad V = \begin{pmatrix} Z_1 & 0 \\ -\omega & Z_2 \end{pmatrix}$$

According to $R_0 = FV^{-1}$, it is easy to obtain the primary reproduction number $R_0 = \frac{b\beta\omega(d+\delta-dp)}{Z_1 Z_2 \left( dZ_4 + b\sigma(d+\delta-dp) \right)}$ of the system (4).

### 3.1. Stability of disease-free equilibrium point

**Theorem 1** When $R_0 \le 1$, the disease-free equilibrium $P^0$ of system (4) is locally asymptotically stable.

Proof: Jacobian matrix at the disease-free equilibrium point $P^0$ of system (4) is:

$$J(P^0) = \begin{bmatrix} -d & \theta & -\frac{\beta S^0}{1+\sigma S^0} & 0 & \delta \\ 0 & -Z_1 & \frac{\beta S^0}{1+\sigma S^0} & 0 & 0 \\ 0 & \omega & -Z_2 & 0 & 0 \\ 0 & 0 & \varepsilon & -Z_3 & 0 \\ 0 & \xi & \gamma & \eta & -Z_4 \end{bmatrix} \tag{5}$$

The characteristic equation of matrix (5) is:

$$|\lambda E - J(P^0)| = \begin{bmatrix} \lambda - (-d) & -\theta & \dfrac{\beta S^0}{1+\sigma S^0} & 0 & -\delta \\ 0 & \lambda - (-Z_1) & -\dfrac{\beta S^0}{1+\sigma S^0} & 0 & 0 \\ 0 & -\omega & \lambda - (-Z_2) & 0 & 0 \\ 0 & 0 & -\varepsilon & \lambda - (-Z_3) & 0 \\ 0 & -\xi & -\gamma & -\eta & \lambda - (-Z_4) \end{bmatrix} \tag{6}$$

So we can get $(\lambda + d)(\lambda + Z_1)(\lambda + Z_2)(\lambda + Z_3)(\lambda + Z_4) = 0$, and it can be obtained by calculation: $\lambda_1 = -d < 0$, $\lambda_2 = -Z_1 < 0$, $\lambda_3 = -Z_2 < 0$, $\lambda_4 = -Z_3 < 0$, $\lambda_5 = -Z_4 < 0$; that is, the genuine part of all characteristic roots of equation (6) is negative, so when $R_0 \leq 1$, the disease-free equilibrium point $P^0$ is locally asymptotically stable, and Theorem 1 is proved.

### 3.2. Stability of positive equilibrium point

When $R_0 > 1$, the system (4) is linearized as:

$$\begin{cases} \dfrac{dS(t)}{dt} = l_{11}S(t) + m_{11}S(t-\tau_1) + l_{12}E(t) + m_{13}I(t-\tau_1) + n_{15}R(t-\tau_2), \\ \dfrac{dE(t)}{dt} = m_{21}S(t-\tau_1) + l_{22}E(t) + m_{23}I(t-\tau_1), \\ \dfrac{dI(t)}{dt} = l_{32}E(t) + l_{33}I(t), \\ \dfrac{dQ(t)}{dt} = l_{43}I(t) + l_{44}Q(t), \\ \dfrac{dR(t)}{dt} = l_{52}E(t) + l_{53}I(t) + l_{54}Q(t) + l_{55}R(t) + n_{55}R(t-\tau_2), \end{cases} \tag{7}$$

where $l_{11} = -d$, $m_{11} = -\dfrac{\beta I^*}{(1+\sigma S^*)^2}$, $l_{12} = \theta$, $m_{13} = -\dfrac{\beta S^*}{1+\sigma S^*}$, $n_{15} = \delta$, $m_{21} = \dfrac{\beta I^*}{(1+\sigma S^*)^2}$, $l_{22} = -Z_1$, $m_{23} = \dfrac{\beta S^*}{1+\sigma S^*}$, $l_{32} = \omega$, $l_{33} = -Z_2$, $l_{43} = \varepsilon$, $l_{44} = -Z_3$, $l_{52} = \xi$, $l_{53} = \gamma$, $l_{54} = \eta$, $l_{55} = d$, $n_{55} = -\delta$; Then the Jacobian matrix at the positive equilibrium point $P^*$ of system (7) is:

$$J(P^*) = \begin{bmatrix} l_{11} + m_{11}e^{-\lambda\tau_1} & l_{12} & m_{13}e^{-\lambda\tau_1} & 0 & n_{15}e^{-\lambda\tau_2} \\ m_{21}e^{-\lambda\tau_1} & l_{22} & m_{23}e^{-\lambda\tau_1} & 0 & 0 \\ 0 & l_{32} & l_{33} & 0 & 0 \\ 0 & 0 & l_{43} & l_{44} & 0 \\ 0 & l_{52} & l_{53} & l_{54} & l_{55} + n_{55}e^{-\lambda\tau_2} \end{bmatrix} \tag{8}$$

The characteristic equation of matrix (8) is:

$$(\lambda - l_{11} - m_{11}e^{-\lambda\tau_1})(\lambda - l_{22})(\lambda - l_{33})(\lambda - l_{44})(\lambda - l_{55} + n_{55}e^{-\lambda\tau_2}) - l_{32}l_{43}l_{54}m_{21}n_{15}e^{-\lambda(\tau_1+\tau_2)} = 0 \tag{9}$$

According to equation (9):

$$X_0(\lambda) + X_1(\lambda)e^{-\lambda\tau_1} + X_2(\lambda)e^{-\lambda\tau_2} + X_3(\lambda)e^{-\lambda(\tau_1+\tau_2)} = 0 \tag{10}$$

where
$X_0(\lambda) = \lambda^5 + A_4\lambda^4 + A_3\lambda^3 + A_2\lambda^2 + A_1\lambda + A_0$, $X_1(\lambda) = B_4\lambda^4 + B_3\lambda^3 + B_2\lambda^2 + B_1\lambda + B_0$,
$X_2(\lambda) = C_4\lambda^4 + C_3\lambda^3 + C_2\lambda^2 + C_1\lambda + C_0$, $X_3(\lambda) = D_3\lambda^3 + D_2\lambda^2 + D_1\lambda + D_0$,
$A_4 = -(l_{11} + l_{22} + l_{33} + l_{44} + l_{55})$, $A_3 = l_{11}l_{55} + l_{11}l_{22} + l_{11}l_{33} + l_{11}l_{44} + l_{22}l_{55} + l_{33}l_{55} + l_{44}l_{55} + l_{22}l_{33} + l_{22}l_{44} + l_{33}l_{44}$,
$A_2 = -(l_{11}l_{22}l_{55} + l_{11}l_{33}l_{55} + l_{11}l_{44}l_{55} + l_{11}l_{22}l_{33} + l_{11}l_{22}l_{44} + l_{11}l_{33}l_{44} + l_{22}l_{33}l_{55} + l_{22}l_{44}l_{55} + l_{33}l_{44}l_{55} - l_{22}l_{33}l_{44})$,
$A_1 = l_{11}l_{22}l_{33}l_{55} + l_{11}l_{22}l_{44}l_{55} + l_{11}l_{44}l_{33}l_{55} - l_{11}l_{22}l_{33}l_{44} - l_{44}l_{22}l_{33}l_{55}$, $A_0 = l_{11}l_{22}l_{33}l_{44}l_{55}$, $B_4 = -m_{11}$,

$B_3 = l_{22}m_{11} + l_{33}m_{11} + l_{44}m_{11} + l_{55}m_{11},\ B_1 = l_{22}l_{33}l_{55}m_{11} + l_{22}l_{44}l_{55}m_{11} + l_{33}l_{44}l_{55}m_{11} - l_{22}l_{33}l_{44}m_{11},$

$B_2 = -(l_{22}l_{55}m_{11} + l_{33}l_{55}m_{11} + l_{44}l_{55}m_{11} + l_{22}l_{33}m_{11} + l_{22}l_{44}m_{11} + l_{33}l_{44}m_{11}),\ B_0 = l_{22}l_{33}l_{44}l_{55}m_{11},\ C_4 = -n_{55},$

$C_2 = -(l_{11}l_{22}n_{55} + l_{11}l_{33}n_{55} + l_{11}l_{44}n_{55} + l_{22}l_{33}n_{55} + l_{22}l_{44}n_{55} + l_{33}l_{44}n_{55}),\ C_0 = l_{11}l_{22}l_{33}l_{44}n_{55},$

$C_3 = l_{11}n_{55} + l_{22}n_{55} + l_{33}n_{55} + l_{44}n_{55},\ C_1 = l_{11}l_{22}l_{33}n_{55} + l_{11}l_{22}l_{44}n_{55} + l_{11}l_{33}l_{44}n_{55} - l_{22}l_{33}l_{44}n_{55},\ D_3 = m_{11}n_{55},$

$D_2 = -(l_{22}m_{11}n_{55} + l_{33}m_{11}n_{55} + l_{44}m_{11}n_{55}),\ D_1 = l_{22}l_{33}m_{11}n_{55} + l_{22}l_{44}m_{11}n_{55} + l_{33}l_{44}m_{11}n_{55},\ D_0 = l_{22}l_{33}l_{44}m_{11}n_{55}$

Next, we will discuss the stability and Hopf bifurcation of positive equilibrium points in different states.

**Case 1** When $\tau_1 = \tau_2 = 0$, equation (10) can be reduced to:

$$\lambda^5 + A_4\lambda^4 + A_3\lambda^3 + A_2\lambda^2 + A_1\lambda + A_0 = 0 \tag{11}$$

**Theorem 2** According to Routh-Hurwitz criterion, if the assumption condition (R) holds, when $\tau_1 = \tau_2 = 0$, the positive equilibrium point $P^*$ is locally asymptotically stable.

$$(R) = \begin{cases} A_3A_4 - A_2 > 0 \\ A_2\left(A_3A_4 - A_2\right) - A_4\left(A_1A_4 - A_0\right) > 0 \\ \left[A_2\left(A_3A_4 - A_2\right) - A_4\left(A_1A_4 - A_0\right)\right]\left(A_1A_4 - A_0\right) > A_0(A_3A_4 - A_2)^2 \end{cases}$$

**Case 2** When $\tau_1 > 0,\ \tau_2 = 0$, equation (10) can be reduced to:

$$X_0(\lambda) + X_2(\lambda) + (X_1(\lambda) + X_3(\lambda))\,e^{-\lambda\tau_1} = 0 \tag{12}$$

Suppose $\lambda = i\varpi_1$ is a root of equation (12), then:

$$\begin{cases} f_{11}(\varpi_1)\cos\varpi_1\tau_1 + h_{11}(\varpi_1)\sin\varpi_1\tau_1 = g_{11}(\varpi_1) \\ f_{12}(\varpi_1)\cos\varpi_1\tau_1 + h_{12}(\varpi_1)\sin\varpi_1\tau_1 = g_{12}(\varpi_1) \end{cases} \tag{13}$$

Here

$f_{11}(\varpi_1) = B_4\varpi_1^4 - B_2\varpi_1^2 + B_0 - D_2\varpi_1^2 + D_0 \qquad f_{12}(\varpi_1) = -B_3\varpi_1^3 + B_1\varpi_1 - D_3\varpi_1^3 - D_1\varpi_1$

$h_{11}(\varpi_1) = -B_3\varpi_1^3 + B_1\varpi_1 - D_3\varpi_1^3 + D_1\varpi_1 \qquad h_{12}(\varpi_1) = -B_4\varpi_1^4 + B_2\varpi_1^2 - B_0 + D_2\varpi_1^2 - D_0$

$g_{11}(\varpi_1) = -A_4\varpi_1^4 + A_2\varpi_1^2 - A_0 - C_4\varpi_1^4 + C_2\varpi_1^2 - C_0 \quad g_{12}(\varpi_1) = -\varpi_1^5 + A_3\varpi_1^3 - A_1\varpi_1 + C_3\varpi_1^3 - C_1\varpi_1$

It can be obtained from (13):
$$\begin{cases} \sin\varpi_1\tau_1 = \dfrac{g_{11}(\varpi_1)f_{12}(\varpi_1) - g_{12}(\varpi_1)f_{11}(\varpi_1)}{h_{11}(\varpi_1)f_{12}(\varpi_1) - h_{12}(\varpi_1)f_{11}(\varpi_1)} \\ \cos\varpi_1\tau_1 = \dfrac{g_{11}(\varpi_1)h_{12}(\varpi_1) - g_{12}(\varpi_1)h_{11}(\varpi_1)}{h_{12}(\varpi_1)f_{11}(\varpi_1) - h_{11}(\varpi_1)f_{12}(\varpi_1)} \end{cases}$$

According to the formula of the sum of squares of the trigonometric function: $\sin^2\varpi_1\tau_1 + \cos^2\varpi_1\tau_1 = 1$, we can obtain the following:

$$\begin{aligned} \left[h_{12}(\varpi_1)f_{11}(\varpi_1) - h_{11}(\varpi_1)f_{12}(\varpi_1)\right]^2 = {} & \left[g_{11}(\varpi_1)f_{12}(\varpi_1) - g_{12}(\varpi_1)f_{11}(\varpi_1)\right]^2 \\ & + \left[g_{11}(\varpi_1)h_{12}(\varpi_1) - g_{12}(\varpi_1)h_{11}(\varpi_1)\right]^2 \end{aligned} \tag{14}$$

Assuming that $\varpi_{11}$ is a positive root of equation (14), then:

$$\tau_1^{(k)} = \frac{1}{\varpi_{11}} * \left[\arccos\frac{g_{11}(\varpi_{11})h_{12}(\varpi_{11}) - g_{12}(\varpi_{11})h_{11}(\varpi_{11})}{h_{12}(\varpi_{11})f_{11}(\varpi_{11}) - h_{11}(\varpi_{11})f_{12}(\varpi_{11})} + 2k\pi\right], k = 0, 1, 2\cdots; \tag{15}$$

$$\tau_{11} = \min\left\{\tau_1^{(k)}, k = 0, 1, 2, \cdots\right\}$$

For equation (12), taking the derivative of $\lambda$ for $\tau_1$ gives:

$$\left(\frac{d\lambda}{d\tau_1}\right)^{-1} = -\frac{X_0' + X_2'}{\lambda(X_0 + X_2)} + \frac{X_1' + X_3'}{\lambda(X_1 + X_3)} - \frac{\tau_1}{\lambda} \tag{16}$$

Therefore:

$$\text{Re}\left(\frac{d\lambda}{d\tau_1}\right)^{-1}_{\lambda=i\varpi_1} = \frac{P_{11}Q_{11} + P_{12}Q_{12}}{Q_{11}^2 + Q_{12}^2} - \frac{P_{13}Q_{13} + P_{14}Q_{14}}{Q_{13}^2 + Q_{14}^2} \tag{17}$$

With

$$P_{11} = -3B_3\varpi_1^2 + B_1 - 3D_3\varpi_1^2 + D_1 \qquad P_{12} = -4B_4\varpi_1^3 + 2B_2\varpi_1 + 2D_2\varpi_1$$
$$P_{13} = 5\varpi_1^4 - 3A_3\varpi_1^2 + A_1 - 3C_3\varpi_1^2 + C_1 \qquad P_{14} = -4A_4\varpi_1^3 + 2A_2\varpi_1 - 4C_4\varpi_1^2 + 2C_2\varpi_1$$
$$Q_{11} = B_3\varpi_1^4 - B_1\varpi_1^2 + D_3\varpi_1^4 - D_1\varpi_1^2 \qquad Q_{12} = B_4\varpi_1^5 - B_2\varpi_1^3 + B_0\varpi_1 - D_2\varpi_1^3 + D_0\varpi_1$$
$$Q_{13} = 4A_4\varpi_1^4 - 2A_2\varpi_1^2 + 4C_4\varpi_1^4 - 2C_2\varpi_1^2 \qquad Q_{14} = 5\varpi_1^5 - 3A_3\varpi_1^3 + A_1\varpi_1 - 3C_3\varpi_1^3 + C_1\varpi_1$$

**Theorem 3** If $R_0 > 1$, according to Hopf bifurcation theory [31]: When $\tau_1 \in [0, \tau_{11}]$, $\tau_2 = 0$, if $\text{Re}\left(\frac{d\lambda}{d\tau_1}\right)^{-1}_{\lambda=i\varpi_1} \neq 0$ is true, then the positive equilibrium point $P^*\left(S^*, E^*, I^*, Q^*, R^*\right)$ is locally asymptotically stable; when $\tau_1 > \tau_{11}$, the positive equilibrium point $P^*\left(S^*, E^*, I^*, Q^*, R^*\right)$ is unstable, and Hopf bifurcation is generated.

**Case 3** When $\tau_1 = 0$, $\tau_2 > 0$, equation (10) can be reduced to:

$$X_0(\lambda) + X_1(\lambda) + (X_2(\lambda) + X_3(\lambda))\, e^{-\lambda\tau_2} = 0 \tag{18}$$

Suppose that $\lambda = i\varpi_2$ is a root of equation (18), then:

$$\begin{cases} f_{21}(\varpi_2)\cos\varpi_2\tau_2 + h_{21}(\varpi_2)\sin\varpi_2\tau_2 = g_{21}(\varpi_2) \\ f_{22}(\varpi_2)\cos\varpi_2\tau_2 + h_{22}(\varpi_2)\sin\varpi_2\tau_2 = g_{22}(\varpi_2) \end{cases} \tag{19}$$

With

$$f_{21}(\varpi_2) = C_4\varpi_2^4 - C_2\varpi_2^2 + C_0 - D_2\varpi_2^2 + D_0 \qquad f_{22}(\varpi_2) = -C_3\varpi_2^3 + C_1\varpi_2 - D_3\varpi_2^3 + D_1\varpi_2$$
$$h_{21}(\varpi_2) = -C_3\varpi_2^3 + C_1\varpi_2 - D_3\varpi_2^3 - D_1\varpi_2 \qquad h_{22}(\varpi_2) = -C_4\varpi_2^4 + C_2\varpi_2^2 - C_0 + D_2\varpi_2^2 - D_0$$
$$g_{21}(\varpi_2) = -A_4\varpi_2^4 + A_2\varpi_2^2 - A_0 - B_4\varpi_2^4 + B_2\varpi_2^2 - B_0 \quad g_{22}(\varpi_2) = -\varpi_2^5 + A_3\varpi_2^3 - A_1\varpi_2 + B_3\varpi_2^3 - B_1\varpi_2$$

It can be obtained from (19): $\begin{cases} \sin\varpi_2\tau_2 = \dfrac{g_{21}(\varpi_2)f_{22}(\varpi_2) - g_{22}(\varpi_2)f_{21}(\varpi_2)}{h_{21}(\varpi_2)f_{22}(\varpi_2) - h_{22}(\varpi_2)f_{21}(\varpi_2)} \\ \cos\varpi_2\tau_2 = \dfrac{g_{21}(\varpi_2)h_{22}(\varpi_2) - g_{22}(\varpi_2)h_{21}(\varpi_2)}{h_{22}(\varpi_2)f_{21}(\varpi_2) - h_{21}(\varpi_2)f_{22}(\varpi_2)} \end{cases}$

According to the formula of the sum of squares of the trigonometric function: $\sin^2\varpi_1\tau_1 + \cos^2\varpi_1\tau_1 = 1$, we can obtain the following:

$$[h_{22}(\varpi_2)f_{21}(\varpi_2) - h_{21}(\varpi_2)f_{22}(\varpi_2)]^2 = [g_{21}(\varpi_2)f_{22}(\varpi_2) - g_{22}(\varpi_1)f_{21}(\varpi_2)]^2$$
$$+ [g_{21}(\varpi_2)h_{22}(\varpi_2) - g_{22}(\varpi_2)h_{21}(\varpi_2)]^2 \tag{20}$$

Assuming that $\varpi_{22}$ is a positive root of equation (20), then:

$$\tau_2^{(k)} = \frac{1}{\varpi_{22}} * \left[ \arccos \frac{g_{21}(\varpi_{22})h_{22}(\varpi_{22}) - g_{22}(\varpi_{22})h_{21}(\varpi_{22})}{h_{22}(\varpi_{22})f_{21}(\varpi_{22}) - h_{21}(\varpi_{22})f_{22}(\varpi_{22})} + 2k\pi \right], k = 0, 1, 2 \cdots ; \tag{21}$$

$\tau_{22} = \min \left\{ \tau_2^{(k)}, k = 0, 1, 2, \cdots \right\}$

For equation (18), taking the derivative of $\lambda$ for $\tau_2$ gives:

$$\left( \frac{d\lambda}{d\tau_2} \right)^{-1} = -\frac{X_0' + X_1'}{\lambda (X_0 + X_1)} + \frac{X_2' + X_3'}{\lambda (X_2 + X_3)} - \frac{\tau_2}{\lambda} \tag{22}$$

Therefore:

$$\mathrm{Re} \left( \frac{d\lambda}{d\tau_2} \right)^{-1}_{\lambda = i\varpi_2} = \frac{P_{21}Q_{21} + P_{22}Q_{22}}{Q_{21}^2 + Q_{22}^2} - \frac{P_{23}Q_{23} + P_{24}Q_{24}}{Q_{23}^2 + Q_{24}^2} \tag{23}$$

With

$P_{21} = -3C_3\varpi_2^2 + C_1 - 3D_3\varpi_2^2 + D_1$  $\qquad P_{22} = -4C_4\varpi_2^3 + 2C_2\varpi_2 + 2D_2\varpi_2$

$P_{23} = 5\varpi_2^4 - 3A_3\varpi_2^2 + A_1 - 3B_3\varpi_2^2 + B_1$ $\qquad P_{24} = -4A_4\varpi_2^3 + 2A_2\varpi_2 - 4B_4\varpi_2^2 + 2B_2\varpi_2$

$Q_{21} = C_3\varpi_2^4 - C_1\varpi_2^2 + D_3\varpi_2^4 - D_1\varpi_2^2$ $\qquad Q_{22} = C_4\varpi_2^5 - C_2\varpi_2^3 + C_0\varpi_2 - D_2\varpi_2^3 + D_0\varpi_2$

$Q_{23} = -\varpi_2^6 + A_3\varpi_2^4 - A_1\varpi_2^2 + B_3\varpi_2^4 + B_1\varpi_2^2$ $\qquad Q_{24} = A_4\varpi_2^5 - A_2\varpi_2^3 + A_0\varpi_2 + B_4\varpi_2^5 - B_2\varpi_2^3 + B_0\varpi_2$

**Theorem 4** If $R_0 > 1$, according to Hopf bifurcation theory [31]: When $\tau_1 = 0$, $\tau_2 \in [0, \tau_{22}]$, if $\mathrm{Re} \left( \frac{d\lambda}{d\tau_2} \right)^{-1}_{\lambda = i\varpi_2} \neq 0$ is true, then the positive equilibrium point $P^* (S^*, E^*, I^*, Q^*, R^*)$ is locally asymptotically stable; when $\tau_2 > \tau_{22}$, the positive equilibrium point $P^* (S^*, E^*, I^*, Q^*, R^*)$ is unstable, and Hopf bifurcation is generated.

**Case 4** When $\tau_1 = \tau_2 = \tau_*$, equation (10) can be reduced to:

$$X_0(\lambda) + [X_1(\lambda) + X_2(\lambda)] e^{-\lambda\tau_*} + X_3(\lambda)e^{-2\lambda\tau_*} = 0 \tag{24}$$

Multiply both sides of the equation by $e^{\lambda\tau_*}$, and we get:

$$X_0(\lambda)e^{\lambda\tau_*} + X_1(\lambda) + X_2(\lambda) + X_3(\lambda)e^{-\lambda\tau_*} = 0 \tag{25}$$

Suppose that $\lambda = i\varpi_*$ is a root of equation (25), then:

$$\begin{cases} f_{31}(\varpi_*) \cos \varpi_*\tau_* + h_{31}(\varpi_*) \sin \varpi_*\tau_* = g_{31}(\varpi_*) \\ f_{32}(\varpi_*) \cos \varpi_*\tau_* + h_{32}(\varpi_*) \sin \varpi_*\tau_* = g_{32}(\varpi_*) \end{cases} \tag{26}$$

With

$f_{31}(\varpi_*) = A_4\varpi_*^4 - A_2\varpi_*^2 + A_0 - D_2\varpi_*^2 + D_0$ $\qquad f_{32}(\varpi_*) = \varpi_*^5 - A_3\varpi_*^3 + A_1\varpi_* - D_3\varpi_*^3 + D_1\varpi_*$

$h_{31}(\varpi_*) = -\varpi_*^5 + A_3\varpi_*^3 - A_1\varpi_* + D_3\varpi_*^3 - D_1\varpi_*$ $\qquad h_{32}(\varpi_*) = A_4\varpi_*^4 - A_2\varpi_*^2 + A_0 + D_2\varpi_*^2 - D_0$

$g_{31}(\varpi_*) = -B_4\varpi_*^4 + B_2\varpi_*^2 - B_0 - C_4\varpi_*^4 + C_2\varpi_*^2 - C_0$ $\qquad g_{32}(\varpi_*) = B_3\varpi_*^3 - B_1\varpi_* + C_3\varpi_*^3 - C_1\varpi_*$

It can be obtained from (26): $\begin{cases} \sin \varpi_*\tau_* = \dfrac{g_{31}(\varpi_*)f_{32}(\varpi_*) - g_{32}(\varpi_*)f_{31}(\varpi_*)}{h_{31}(\varpi_*)f_{32}(\varpi_*) - h_{32}(\varpi_*)f_{31}(\varpi_*)} \\ \cos \varpi_*\tau_* = \dfrac{g_{31}(\varpi_*)h_{32}(\varpi_*) - g_{32}(\varpi_*)h_{31}(\varpi_*)}{h_{32}(\varpi_*)f_{31}(\varpi_*) - h_{31}(\varpi_*)f_{32}(\varpi_*)} \end{cases}$

According to the formula of the sum of squares of the trigonometric function: $\sin^2 \varpi_1 \tau_1 + \cos^2 \varpi_1 \tau_1 = 1$, we can obtain the following:

$$\left[h_{32}(\varpi_*)f_{31}(\varpi_*) - h_{31}(\varpi_*)f_{32}(\varpi_*)\right]^2 = \left[g_{31}(\varpi_*)f_{31}(\varpi_*) - g_{32}(\varpi_*)f_{31}(\varpi_*)\right]^2 \\ + \left[g_{31}(\varpi_*)h_{32}(\varpi_*) - g_{32}(\varpi_*)h_{31}(\varpi_*)\right]^2 \tag{27}$$

Assuming that $\varpi_{**}$ is a positive root of equation (27), then:

$$\tau_*^{(k)} = \frac{1}{\varpi_{**}} * \left[\arccos \frac{g_{31}(\varpi_{**})h_{32}(\varpi_{**}) - g_{32}(\varpi_{**})h_{31}(\varpi_{**})}{h_{32}(\varpi_{**})f_{31}(\varpi_{**}) - h_{31}(\varpi_{**})f_{32}(\varpi_{**})} + 2k\pi\right], k = 0, 1, 2 \cdots ; \tag{28}$$

$\tau_{**} = \min\left\{\tau_*^{(k)}, k = 0, 1, 2, \cdots\right\}$

For equation (25), taking the derivative of $\lambda$ for $\tau_*$ gives:

$$\left(\frac{d\lambda}{d\tau_*}\right)^{-1} = -\frac{X_0' e^{\lambda\tau_*} + X_1' + X_2' + X_3' e^{-\lambda\tau_*}}{\lambda X_0 e^{-\lambda\tau_*} + \lambda X_3 e^{\lambda\tau_*}} - \frac{\tau_*}{\lambda} \tag{29}$$

Therefore:

$$\mathrm{Re}\left(\frac{d\lambda}{d\tau_*}\right)^{-1}_{\lambda=i\varpi_*} = \frac{P_{31}Q_{31} + P_{32}Q_{32}}{Q_{31}^2 + Q_{32}^2} \tag{30}$$

With

$P_{31} = -3B_3\varpi_*^2 + B_1 - 3C_3\varpi_*^2 + C_1$
$\quad + \left(5\varpi_*^4 - 3A_3\varpi_*^2 + A_1 - 3D_3\varpi_*^2 + D_1\right)\cos\varpi_*\tau_* + \left(4A_4\varpi_*^3 - 2A_2\varpi_* + 2D_2\varpi_2\right)\sin\varpi_*\tau_*$

$P_{32} = -4B_4\varpi_*^3 + 2B_2\varpi_* - 4C_4\varpi_*^3 + 2C_2\varpi_*$
$\quad + \left(-4A_4\varpi_*^3 + 2A_2\varpi_* + 2D_2\varpi_*\right)\cos\varpi_*\tau_* + \left(5\varpi_*^4 - 3A_3\varpi_*^2 + A_1 + 3D_3\varpi_*^2 - D_1\right)\sin\varpi_*\tau_*$

$Q_{31} = \left(-\varpi_*^6 + A_3\varpi_*^4 - A_1\varpi_*^2 - D_3\varpi_*^4 + D_1\varpi_*^2\right)\cos\varpi_*\tau_* + \left(A_4\varpi_*^5 - A_2\varpi_*^3 + A_0\varpi_* - D_2\varpi_*^2 + D_0\varpi_*\right)\sin\varpi_*\tau_*$

$Q_{31} = \left(\varpi_*^6 - A_3\varpi_*^4 + A_1\varpi_*^2 - D_3\varpi_*^4 + D_1\varpi_*^2\right)\sin\varpi_*\tau_* + \left(A_4\varpi_*^5 - A_2\varpi_*^3 + A_0\varpi_* + D_2\varpi_*^2 - D_0\varpi_*\right)\cos\varpi_*\tau_*,$

**Theorem 5** If $R_0 > 1$, according to Hopf bifurcation theory [31]: When $\tau_1 = \tau_2 = \tau_* < \tau_{**}$, if $\mathrm{Re}\left(\frac{d\lambda}{d\tau_*}\right)^{-1}_{\lambda=i\varpi_*} \neq 0$ is true, then the positive equilibrium point $P^*(S^*, E^*, I^*, Q^*, R^*)$ is locally asymptotically stable; when $\tau_* > \tau_{**}$, the positive equilibrium point $P^*(S^*, E^*, I^*, Q^*, R^*)$ is unstable, and Hopf bifurcation is generated.

**Case 5** When $\tau_1 > 0$, $\tau_2 \in [0, \tau_{22})$, this situation is similar to case 2, equation (10) can be reduced to:

$$X_0(\lambda) + X_2(\lambda) + (X_1(\lambda) + X_3(\lambda))e^{-\lambda\tau_1} = 0 \tag{31}$$

Suppose that $\lambda = i\varpi_3$ is a root of equation (31), then:

$$\begin{cases} f_{41}(\varpi_3)\cos\varpi_3\tau_3 + h_{41}(\varpi_1)\sin\varpi_3\tau_3 = g_{41}(\varpi_3) \\ f_{42}(\varpi_3)\cos\varpi_3\tau_3 + h_{42}(\varpi_3)\sin\varpi_3\tau_3 = g_{42}(\varpi_3) \end{cases} \tag{32}$$

With

$$f_{41}(\varpi_3) = B_4\varpi_3^4 - B_2\varpi_3^2 + B_0 - D_2\varpi_3^2 + D_0 \qquad f_{42}(\varpi_3) = -B_3\varpi_3^3 + B_1\varpi_3 - D_3\varpi_3^3 - D_1\varpi_3$$

$$h_{41}(\varpi_3) = -B_3\varpi_3^3 + B_1\varpi_3 - D_3\varpi_3^3 + D_1\varpi_3 \qquad h_{42}(\varpi_3) = -B_4\varpi_3^4 + B_2\varpi_3^2 - B_0 + D_2\varpi_3^2 - D_0$$

$$g_{41}(\varpi_3) = -A_4\varpi_3^4 + A_2\varpi_3^2 - A_0 - C_4\varpi_3^4 + C_2\varpi_3^2 - C_0 \qquad g_{42}(\varpi_3) = -\varpi_3^5 + A_3\varpi_3^3 - A_1\varpi_3 + C_3\varpi_3^3 - C_1\varpi_3$$

It can be obtained from (32):
$$\begin{cases} \sin\varpi_3\tau_3 = \dfrac{g_{41}(\varpi_3)f_{42}(\varpi_3) - g_{42}(\varpi_3)f_{41}(\varpi_3)}{h_{41}(\varpi_3)f_{42}(\varpi_3) - h_{42}(\varpi_3)f_{41}(\varpi_3)} \\ \cos\varpi_3\tau_3 = \dfrac{g_{41}(\varpi_3)h_{42}(\varpi_3) - g_{42}(\varpi_3)h_{41}(\varpi_3)}{h_{42}(\varpi_3)f_{41}(\varpi_3) - h_{41}(\varpi_3)f_{42}(\varpi_3)} \end{cases}$$

According to the formula of the sum of squares of the trigonometric function: $\sin^2\varpi_1\tau_1 + \cos^2\varpi_1\tau_1 = 1$, we can obtain the following:

$$\begin{aligned}[h_{42}(\varpi_3)f_{41}(\varpi_3) - h_{41}(\varpi_3)f_{42}(\varpi_3)]^2 &= [g_{41}(\varpi_3)f_{42}(\varpi_3) - g_{42}(\varpi_3)f_{41}(\varpi_3)]^2 \\ &\quad + [g_{41}(\varpi_3)h_{42}(\varpi_3) - g_{42}(\varpi_3)h_{41}(\varpi_3)]^2\end{aligned} \tag{33}$$

Assuming that $\varpi_{33}$ is a positive root of equation (33), then:

$$\tau_3^{(k)} = \frac{1}{\varpi_{33}} * \left[ \arccos\frac{g_{41}(\varpi_{33})h_{42}(\varpi_{33}) - g_{42}(\varpi_{33})h_{41}(\varpi_{33})}{h_{42}(\varpi_{33})f_{41}(\varpi_{33}) - h_{41}(\varpi_{33})f_{42}(\varpi_{33})} + 2k\pi \right], k = 0, 1, 2\cdots; \tag{34}$$

$$\tau_{33} = \min\left\{ \tau_3^{(k)}, k = 0, 1, 2, \cdots \right\}$$

For equation (31), taking the derivative of $\lambda$ for $\tau_3$ gives:

$$\left( \frac{d\lambda}{d\tau_3} \right)^{-1} = -\frac{X_0' + X_2'}{\lambda(X_0 + X_2)} + \frac{X_1' + X_3'}{\lambda(X_1 + X_3)} - \frac{\tau_3}{\lambda} \tag{35}$$

Therefore:

$$\text{Re}\left( \frac{d\lambda}{d\tau_3} \right)^{-1}_{\lambda=i\varpi_3} = \frac{P_{41}Q_{41} + P_{42}Q_{42}}{Q_{41}^2 + Q_{42}^2} - \frac{P_{43}Q_{43} + P_{44}Q_{44}}{Q_{43}^2 + Q_{44}^2} \tag{36}$$

With

$$P_{41} = -3B_3\varpi_3^2 + B_1 - 3D_3\varpi_3^2 + D_1 \qquad P_{42} = -4B_4\varpi_3^3 + 2B_2\varpi_3 + 2D_2\varpi_3$$

$$P_{43} = 5\varpi_3^4 - 3A_3\varpi_3^2 + A_1 - 3C_3\varpi_3^2 + C_1 \qquad P_{44} = -4A_4\varpi_3^3 + 2A_2\varpi_3 - 4C_4\varpi_3^2 + 2C_2\varpi_3$$

$$Q_{41} = B_3\varpi_3^4 - B_1\varpi_3^2 + D_3\varpi_3^4 - D_1\varpi_3^2 \qquad Q_{42} = B_4\varpi_3^5 - B_2\varpi_3^3 + B_0\varpi_3 - D_2\varpi_3^3 + D_0\varpi_3$$

$$Q_{43} = 4A_4\varpi_3^4 - 2A_2\varpi_3^2 + 4C_4\varpi_3^4 - 2C_2\varpi_3^2 \qquad Q_{44} = 5\varpi_3^5 - 3A_3\varpi_3^3 + A_1\varpi_3 - 3C_3\varpi_3^3 + C_1\varpi_3$$

**Theorem 6** If $R_0 > 1$, according to Hopf bifurcation theory [31]: When $\tau_1 \in [0, \tau_{33})$, if $\text{Re}\left( \frac{d\lambda}{d\tau_3} \right)^{-1}_{\lambda=i\varpi_3} \neq 0$ is true, then the positive equilibrium point $P^*(S^*, E^*, I^*, Q^*, R^*)$ is locally asymptotically stable; when $\tau_1 > \tau_{33}$, the positive equilibrium point $P^*(S^*, E^*, I^*, Q^*, R^*)$ is unstable, and Hopf bifurcation is generated.

**Case 6** When $\tau_1 \in [0, \tau_{11})$, $\tau_2 > 0$, this situation is similar to case 3, equation (10) can be reduced to:

$$X_0(\lambda) + X_1(\lambda) + (X_2(\lambda) + X_3(\lambda)) e^{-\lambda\tau_2} = 0 \tag{37}$$

Suppose that $\lambda = i\varpi_4$ is a root of equation (37), then:

$$\begin{cases} f_{51}\left(\varpi_4\right)\cos\varpi_4\tau_4 + h_{51}\left(\varpi_4\right)\sin\varpi_4\tau_4 = g_{51}\left(\varpi_4\right) \\ f_{52}\left(\varpi_4\right)\cos\varpi_4\tau_4 + h_{52}\left(\varpi_4\right)\sin\varpi_4\tau_4 = g_{52}\left(\varpi_4\right) \end{cases} \tag{38}$$

With

$$\begin{aligned} &f_{51}\left(\varpi_4\right) = C_4\varpi_4^4 - C_2\varpi_4^2 + C_0 - D_2\varpi_4^2 + D_0 && f_{52}\left(\varpi_4\right) = -C_3\varpi_4^3 + C_1\varpi_4 - D_3\varpi_4^3 + D_1\varpi_4 \\ &h_{51}\left(\varpi_4\right) = -C_3\varpi_4^3 + C_1\varpi_4 - D_3\varpi_4^3 - D_1\varpi_4 && h_{52}\left(\varpi_4\right) = -C_4\varpi_4^4 + C_2\varpi_4^2 - C_0 + D_2\varpi_4^2 - D_0 \\ &g_{51}\left(\varpi_4\right) = -A_4\varpi_4^4 + A_2\varpi_4^2 - A_0 - B_4\varpi_4^4 + B_2\varpi_4^2 - B_0 && g_{52}\left(\varpi_4\right) = -\varpi_4^5 + A_3\varpi_4^3 - A_1\varpi_4 + B_3\varpi_4^3 - B_1\varpi_4 \end{aligned}$$

It can be obtained from (38):
$$\begin{cases} \sin\varpi_4\tau_4 = \dfrac{g_{51}(\varpi_4)f_{52}(\varpi_4) - g_{52}(\varpi_4)f_{51}(\varpi_4)}{h_{51}(\varpi_4)f_{52}(\varpi_4) - h_{52}(\varpi_4)f_{51}(\varpi_4)} \\ \cos\varpi_4\tau_4 = \dfrac{g_{51}(\varpi_4)h_{52}(\varpi_4) - g_{52}(\varpi_4)h_{51}(\varpi_4)}{h_{52}(\varpi_4)f_{51}(\varpi_4) - h_{51}(\varpi_4)f_{52}(\varpi_4)} \end{cases}$$

According to the formula of the sum of squares of the trigonometric function: $\sin^2\varpi_1\tau_1 + \cos^2\varpi_1\tau_1 = 1$, we can obtain the following:

$$\begin{aligned} \left[h_{52}(\varpi_4)f_{51}(\varpi_4) - h_{51}(\varpi_4)f_{52}(\varpi_4)\right]^2 = &\left[g_{51}(\varpi_4)f_{52}(\varpi_4) - g_{52}(\varpi_4)f_{51}(\varpi_4)\right]^2 \\ &+ \left[g_{51}(\varpi_4)h_{52}(\varpi_4) - g_{53}(\varpi_4)h_{51}(\varpi_4)\right]^2 \end{aligned} \tag{39}$$

Assuming that $\varpi_{44}$ is a positive root of equation (39), then:

$$\tau_4^{(k)} = \frac{1}{\varpi_{44}} * \left[\arccos\frac{g_{51}(\varpi_{44})h_{52}(\varpi_{44}) - g_{52}(\varpi_{44})h_{51}(\varpi_{44})}{h_{52}(\varpi_{44})f_{51}(\varpi_{44}) - h_{51}(\varpi_{44})f_{52}(\varpi_{44})} + 2k\pi\right], k = 0, 1, 2\cdots; \tag{40}$$

$\tau_{44} = \min\left\{\tau_4^{(k)}, k = 0, 1, 2, \cdots\right\}$
For equation (37), taking the derivative of $\lambda$ for $\tau_4$ gives:

$$\left(\frac{d\lambda}{d\tau_4}\right)^{-1} = -\frac{X_0' + X_1'}{\lambda\left(X_0 + X_1\right)} + \frac{X_2' + X_3'}{\lambda\left(X_2 + X_3\right)} - \frac{\tau_4}{\lambda} \tag{41}$$

Therefore:

$$\mathrm{Re}\left(\frac{d\lambda}{d\tau_4}\right)^{-1}_{\lambda=i\varpi_4} = \frac{P_{51}Q_{51} + P_{52}Q_{52}}{Q_{51}^2 + Q_{52}^2} - \frac{P_{53}Q_{53} + P_{54}Q_{54}}{Q_{53}^2 + Q_{54}^2} \tag{42}$$

With

$$\begin{aligned} &P_{51} = -3C_3\varpi_4^2 + C_1 - 3D_3\varpi_4^2 + D_1 && P_{52} = -4C_4\varpi_4^3 + 2C_2\varpi_4 + 2D_2\varpi_4 \\ &P_{53} = 5\varpi_4^4 - 3A_3\varpi_4^2 + A_1 - 3B_3\varpi_4^2 + B_1 && P_{54} = -4A_4\varpi_4^3 + 2A_2\varpi_4 - 4B_4\varpi_4^2 + 2B_2\varpi_4 \\ &Q_{51} = C_3\varpi_4^4 - C_1\varpi_4^2 + D_3\varpi_4^4 - D_1\varpi_4^2 && Q_{52} = C_4\varpi_4^5 - C_2\varpi_4^3 + C_0\varpi_4 - D_2\varpi_4^3 + D_0\varpi_4 \\ &Q_{53} = -\varpi_4^6 + A_3\varpi_4^4 - A_1\varpi_4^2 + B_3\varpi_4^4 + B_1\varpi_4^2 && Q_{54} = A_4\varpi_4^5 - A_2\varpi_4^3 + A_0\varpi_4 + B_4\varpi_4^5 - B_2\varpi_4^3 + B_0\varpi_4 \end{aligned}$$

**Theorem 7** If $R_0 > 1$, according to Hopf bifurcation theory [31]: When $\tau_2 \in [0, \tau_{44})$, if $\mathrm{Re}\left(\frac{d\lambda}{d\tau_4}\right)^{-1}_{\lambda=i\varpi_4} \neq 0$ is true, then the positive equilibrium point $P^*\left(S^*, E^*, I^*, Q^*, R^*\right)$ is locally asymptotically stable; when $\tau_2 > \tau_{44}$, the positive equilibrium point $P^*\left(S^*, E^*, I^*, Q^*, R^*\right)$ is unstable, and Hopf bifurcation is generated.

## 4. Optimal control problem

Real-world effective control of the spread of computer viruses typically necessitates sizable control efforts, resulting in significant financial outlays. In this section, an optimal control system based on system (3) is established based on Pontryagin's optimal control theory [32] to control the spread of the virus so that the number of infected nodes reaches the minimum within a predetermined time frame. As shown in the following formula, this paper introduces a time-dependent control variable based on formula (3):

$$
\begin{cases}
\dfrac{dS(t)}{dt} = (1-p)b + \delta R(t-\tau_2) + \theta E(t) - \dfrac{\beta S(t-\tau_1) I(t-\tau_1)}{1+\sigma S(t-\tau_1)} - dS(t), \\
\dfrac{dE(t)}{dt} = \dfrac{\beta S(t-\tau_1) I(t-\tau_1)}{1+\sigma S(t-\tau_1)} - (\theta + d + \omega + \mu(t))E(t), \\
\dfrac{dI(t)}{dt} = \omega E(t) - (d + \alpha_1 + \mu(t) + \varepsilon)I(t), \\
\dfrac{dQ(t)}{dt} = \varepsilon I(t) - (d + \alpha_2 + \mu(t))Q(t), \\
\dfrac{dR(t)}{dt} = pb + \mu(t)E(t) + \mu(t)I(t) + \mu(t)Q(t) - dR(t) - \delta R(t-\tau_2),
\end{cases}
\tag{43}
$$

Here, the variables $\xi$, $\gamma$ and $\eta$ are replaced by the control variable $\mu(t)$. By adjusting the control variable, such as the effectiveness parameters of antivirus software (including enhancing the detection and removal capabilities of antivirus software, reducing the growth rate of infected nodes, increasing the recovery rate, etc.), the dynamic behavior of the system can be modified. This allows the recovery rate of computers in the latent state e, infected state i, and isolated state q to become time-varying, represented by the control variable $\mu(t)$.

The goal of implementing optimal control is to minimize the number of infected individuals. However, there is a specific cost associated with putting any control strategy into action. As a result, the objective function is defined as follows when taking into account the treatment cost and control cost:

$$
J(\mu) = \int_{t_0}^{t_f} \left( MI(t) + \frac{N}{2}\mu^2(t) \right) dt
\tag{44}
$$

Here, $M$ is the weight coefficient of the infected node; $N$ is the weight coefficient of the control strategy; $\frac{N}{2}\mu^2$ is the cost of the corresponding control strategy. The optimal control problem is to find the optimal control $\mu^*$ so that the objective function $J(\mu)$ take the minimum value, that is, $J(\mu^*) =_U^{\min} J(\mu)$, where $U = \{\mu | 0 \le a \le \mu \le b \le 1\}$ is the control set.

**Theorem 8** For a state system with a known initial value (43) and a given objective function (44), there must exist the following optimal control $\mu^* \in U$, making $J(\mu^*) =_U^{\min} J(\mu)$.

**Proof:** According to the optimal existence theory [33], it is only necessary to prove:

a) The set of control variables and corresponding state variables is non-empty;

b) The control set is a convex closed set;

c) The right side of the state system is bounded for state variables and control variables;

d) The integrand in the objective function is a convex function for the permissible control;

e) There are constants $K_1, K_2 > 0$, and $\Lambda > 1$ so that the integrand in the objective function has an upper bound

$K_1 + K_2 \left( |\mu|^2 \right)^{\frac{\Lambda}{2}}$.

Firstly, conditions (a) and (c) are satisfied according to the previous dynamics discussion of system (3). By definition, the control set is non-empty, and the integrand in the objective function is a convex function for the control set, so the conditions (b) and (d) are satisfied. In addition, because $MI(t) \geq 0$, $MI(t) + \frac{N}{2}\mu^2(t) \geq \frac{N}{2}\mu^2 = \frac{N}{2}\|\vec{\mu}\|^2$, so take $\Lambda = 2$, $K_1 = 0$ and $K_2 = \frac{N}{2}$, condition (e) is established. In summary, Theorem 8 is proven.

According to the Pontryagin minimum principle [32], the necessary conditions for optimal control can be obtained as follows.

The Lagrange function is defined as: $L(\mu) = MI(t) + \frac{N}{2}\mu^2(t)$.

Define the Hamiltonian function $H(S, E, I, Q, R, \mu, \lambda_1, \lambda_2, \lambda_3, \lambda_4, \lambda_5)$ as:

$$
\begin{aligned}
H(t) = \ & MI(t) + \frac{N}{2}\mu^2(t) + \lambda_1 \left\{ (1-p)b + \delta R(t-\tau_2) + \theta E(t) - \frac{\beta S(t-\tau_1)I(t-\tau_1)}{1+\sigma S(t-\tau_1)} - dS(t) \right\} \\
& + \lambda_2 \left\{ \frac{\beta S(t-\tau_1)I(t-\tau_1)}{1+\sigma S(t-\tau_1)} - (\theta + d + \omega + \mu(t))E(t) \right\} \\
& + \lambda_3 \left\{ \omega E(t) - (d + \alpha_1 + \mu(t) + \varepsilon)I(t) \right\} \\
& + \lambda_4 \left\{ \varepsilon I(t) - (d + \alpha_2 + \mu(t))Q(t) \right\} \\
& + \lambda_5 \left\{ pb + \mu(t)E(t) + \mu(t)I(t) + \mu(t)Q(t) - dR(t) - \delta R(t-\tau_2) \right\}
\end{aligned}
$$

**Theorem 9** For the given optimal control $\mu^* \in U$, let the corresponding optimal solution be $S_*, E_*, I_*, Q_*, R_*$, then there is a co-state variable $\lambda_1, \lambda_2, \lambda_3, \lambda_4, \lambda_5$ that satisfies:

a) Costate equation:

$$
\frac{d\lambda_1}{dt} = -\frac{\partial H}{\partial S} = -\left\{ \lambda_1 \left[ \frac{\beta I(t-\tau_1)}{(1+\sigma S(t-\tau_1))^2} - d \right] + \lambda_2 \frac{\beta I(t-\tau_1)}{(1+\sigma S(t-\tau_1))^2} \right\},
$$

$$
\frac{d\lambda_2}{dt} = -\frac{\partial H}{\partial E} = -\left\{ \lambda_1 \theta - \lambda_2 (\theta + d + \omega + \mu(t)) + \lambda_3 \omega \right\},
$$

$$
\frac{d\lambda_3}{dt} = -\frac{\partial H}{\partial I} = -\left\{ M - \lambda_1 \frac{\beta S(t-\tau_1)}{1+\sigma S(t-\tau_1)} + \lambda_2 \frac{\beta S(t-\tau_1)}{1+\sigma S(t-\tau_1)} - \lambda_3 (d + \alpha_1 + \mu(t) + \varepsilon) + \lambda_4 \varepsilon + \lambda_5 \mu(t) \right\},
$$

$$
\frac{d\lambda_4}{dt} = -\frac{\partial H}{\partial Q} = -\left\{ -\lambda_4 (\mu(t) + d + \alpha_2) + \lambda_5 \mu(t) \right\}
$$

$$
\frac{d\lambda_5}{dt} = -\frac{\partial H}{\partial R} = -\left\{ \lambda_1 \delta + \lambda_5 (-d - \delta) \right\}
$$

b) Cross section conditions: $\lambda_1(t_f) = \lambda_2(t_f) = \lambda_3(t_f) = \lambda_4(t_f) = \lambda_5(t_f) = 0$.

c) According to the Pontryagin minimum principle, by solving the partial derivative of the Hamiltonian equation with respect to the control variable, the control point whose derivative is equal to zero can be obtained, and the optimal control solution can be obtained. That means $\frac{\partial H}{\partial \mu} = 0$. And the optimality condition can be obtained:

$$
\mu^* = \max \left\{ a_1, \min \left\{ b_1, \frac{\lambda_2 E(t) + \lambda_3 I(t) + \lambda_4 Q(t) - \lambda_5 (E(t) + I(t) + Q(t))}{N} \right\} \right\}.
$$

This way, a win-win optimal control can be achieved so that the number of infected computers is the least and the cost is the least.

## 5. Numerical simulation

To test the above conclusions, this section revolves around the basic reproduction number $R_0 = \frac{b\beta\omega(d+\delta-dp)}{Z_1 Z_2 (dZ_4 + b\sigma(d+\delta-dp))}$ and conducts numerical simulations based on different parameters. The experimental environment used in this study is MATLAB R2021. All experiments and computational tasks were performed on a computer equipped with an Intel (R) Core (TM) i7-7500U CPU and 8GB RAM.

### 5.1. Stability of disease-free equilibrium point $P^0$

Select parameters: $b = 33$, $p = 0.025$, $d = 0.01$, $\theta = 0.015$, $\beta = 0.0002$, $\sigma = 0.0014$, $\omega = 0.0155$, $\xi = 0.01$, $\varepsilon = 0.014$, $\gamma = 0.03$, $\eta = 0.01$, $\delta = 0.003$, $\alpha_1 = 0.01$, $\alpha_2 = 0.018$, $\tau_1 = 36.9$, $\tau_2 = 35$. Through MATLAB calculation, $R_0 = 0.5612 < 1$ can be obtained. The disease-free equilibrium $P^0(3236.5385, 0, 0, 0, 63.4615)$ exists in system (4). And the disease-free equilibrium point $P^0$ is locally asymptotically stable. The simulation results are shown in Fig 2.

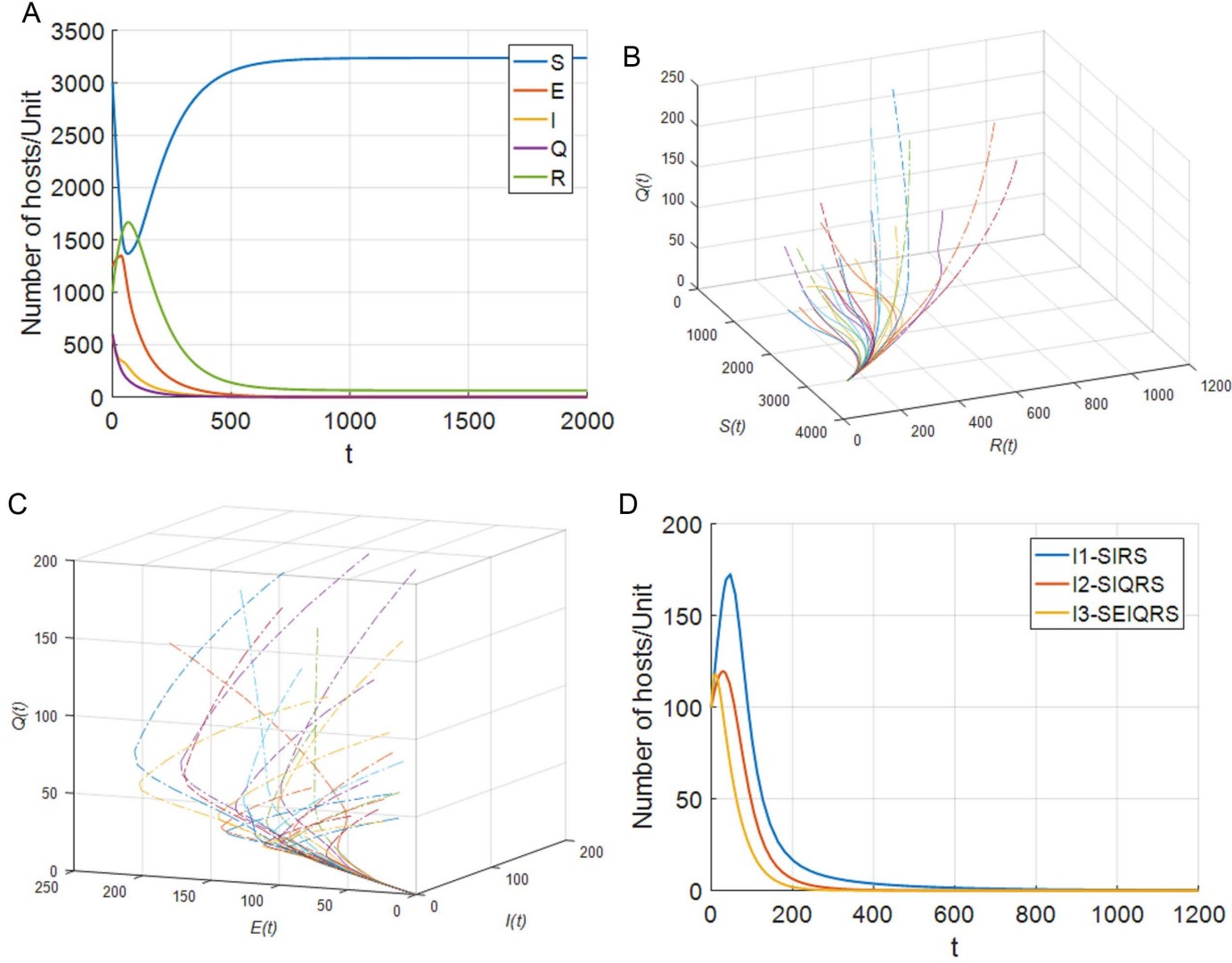

**Fig 2. Number of SEIQRS model hosts when** w (a) The change of each node with respect to time $t$ when $R_0 = 0.5612 < 1$ (b) SQR phase diagram for $R_0 = 0.5612 < 1$ (c) EIQ phase diagram for $R_0 = 0.5612 < 1$ (d) Number of hosts infected with different models when $R_0 < 1$

The phase diagram can provide insights into the dynamics of virus spread, such as the existence of stable equilibrium points, limit cycles, or other patterns of behavior. It allows for visualizing the transitions between different compartments and understanding the long-term behavior of the system. The phase diagram takes different state variables as axes and shows how these variables change over time. By drawing the phase diagram of SQR model, we can observe the transition, equilibrium point and stability characteristics between different states, so as to better understand the dynamic behavior of computer virus propagation.

Fig 2a draws the waveform diagram of each node changing with time $t$ when $R_0 = 0.5612 < 1$, and Fig 2b and 2c illustrate the SQR phase diagram and EIQ phase diagram at this time. As can be seen from the above Fig: when $R_0 = 0.5612 < 1$, no matter how the initial value of hosts in each state changes, over time, the susceptible node $S(t)$ and the immune node $R(t)$ will eventually tend to be stable, while the latent node $E(t)$, the infected node $I(t)$, and the isolated node $Q(t)$ will ultimately converge to 0. Thus, when $R_0 \leq 1$, the disease-free equilibrium point $P^0$ of system (4) tends to be stable. To better verify the effectiveness of model (4), keep the node state transition parameters of V. MadhuSudanan's SIRS model [28] and Fangfang Yang's SIQRS model [29] consistent with the SEIQRS model, let $b = 3$, $\beta = 0.00022$, $\tau_1 = 5$, $\tau_2 = 15$, the changes in the number of infected nodes in each model are shown in Fig 2d. Through comparison, it is found that the result is consistent with Theorem 1, that is, when $R_0 \leq 1$, the system is stable at the disease-free equilibrium point, the virus is eliminated in the computer network, and the infected host will eventually return to normal under anti-virus software. It is also apparent from Fig 2d that the infected node in the SEIQRS model tends to zero the fastest, which also means that the virus can be eliminated faster in the SEIQRS model.

### 5.2. Stability of positive equilibrium $P^*$

Select parameters: $b = 33$, $p = 0.025$, $d = 0.01$, $\theta = 0.0015$, $\beta = 0.0005$, $\sigma = 0.0014$, $\omega = 0.0155$, $\xi = 0.01$, $\varepsilon = 0.014$, $\gamma = 0.03$, $\eta = 0.01$, $\delta = 0.03$, $\alpha_1 = 0.01$, $\alpha_2 = 0.018$. Through MATLAB calculation, $R_0 = 1.9196 > 1$ can be obtained. System (4) has a unique positive equilibrium point $P^*(533.9587, 1254.6701, 303.8654, 111.9504, 590.1792)$. Figs 4–9, respectively, describe the number of hosts of SEIQRS model with different time delays when $R_0 > 1$. In addition, in order to better verify the validity of model (4), keep the node state transition parameters of V. MadhuSudanan's SIRS model [28] and Fangfang Yang's SIQRS model [29] consistent with the SEIQRS model, let $b = 10$, $\tau_1 = 5$, $\tau_2 = 15$, the changes in the number of infected nodes in each model are shown in Fig 3. Through comparison, it is found that when $R_0 > 1$, the system is stable at the positive equilibrium point. It can also be clearly seen from Fig 3 that when the system finally stabilizes at the positive equilibrium point, the number of infected nodes in the SEIQRS model is the least, which also indicates that the computer virus is well controlled in the SEIQRS model.

(1) Fig 4 shows that, when $\tau_1 \in [0, \tau_{11})$, $\tau_2 = 0$, the system (4) is locally asymptotically stable at the positive equilibrium point; Fig 5 shows that when $\tau_1 \geq \tau_{11}$, the system is out of control and unstable and Hopf bifurcation occurs. When $\tau_1 > 0$, $\tau_2 \in [0, \tau_{22})$, case 5 is similar to case 2. When $\tau_1 \in [0, \tau_{33})$, the system is locally asymptotically stable. When $\tau_1 > \tau_{33}$, the system is out of control and produces Hopf bifurcation. The experimental simulation diagrams of those situations are shown in Figs 4 and 5.

(2) Fig 6 shows that, when $\tau_1 = 0$, $\tau_2 \in [0, \tau_{22})$, the system (4) is locally asymptotically stable at the positive equilibrium point; Fig 7 shows that when $\tau_2 \geq \tau_{22}$, the system is out of control and unstable and hopf bifurcation occurs. When $\tau_1 \in [0, \tau_{11})$, $\tau_2 > 0$, case 6 is similar to case 3. When $\tau_2 \in [0, \tau_{44})$, the system is locally asymptotically stable. When $\tau_2 > \tau_{44}$, the system is out of control and produces hopf bifurcation. The experimental simulation diagrams of those situations are shown in Figs 6 and 7.

(3) Fig 8 shows that, when $\tau_1 = \tau_2 < \tau_{**}$, the system (4) is locally asymptotically stable at the positive equilibrium point; Fig 9 shows that when $\tau_1 = \tau_2 > \tau_{**}$, the system is out of control and unstable and Hopf bifurcation occurs.

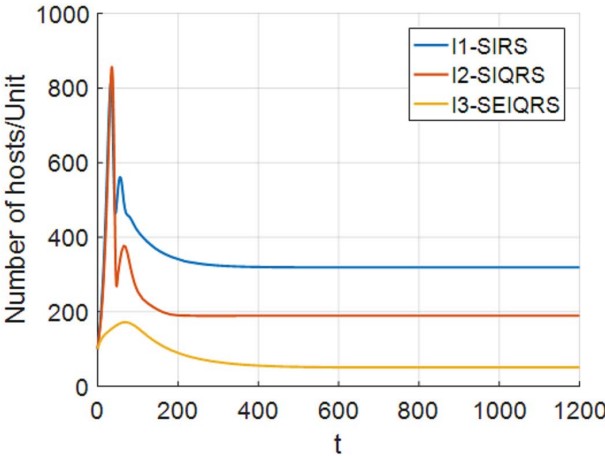

**Fig 3. Number of infected hosts in different models when** $R_0 > 1$

A

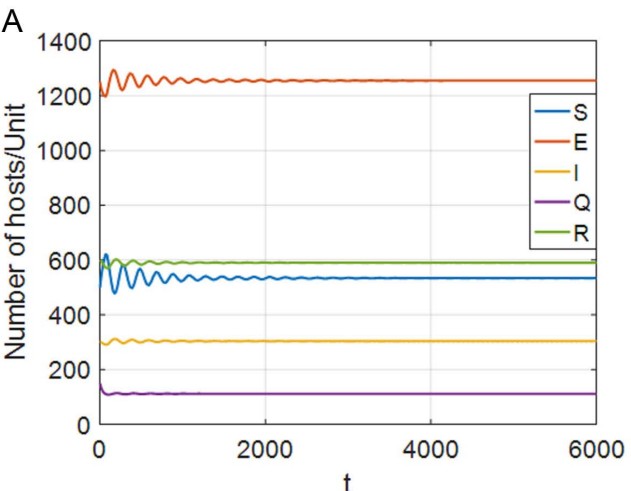

B

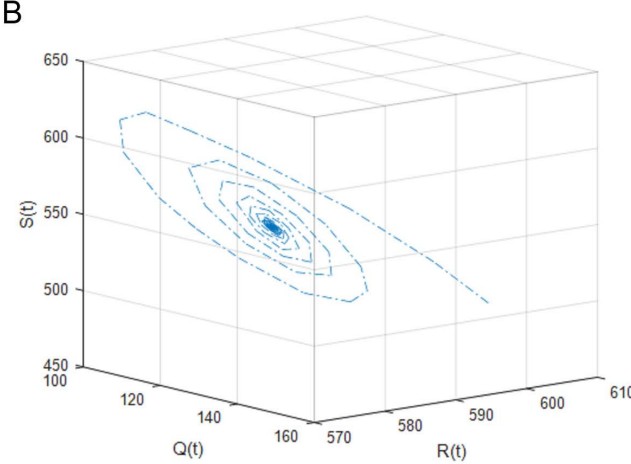

**Fig 4. When** $\tau_1 = 70 < \tau_{11} = 76, \tau_2 = 0$**, the positive equilibrium** $P^*$ **is locally stable.** (a) $S, E, I, Q, R$ with respect to time $t$ (b) SQR phase diagram.

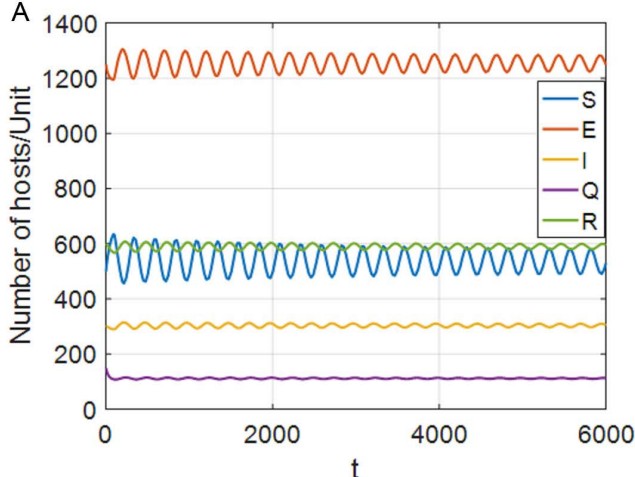

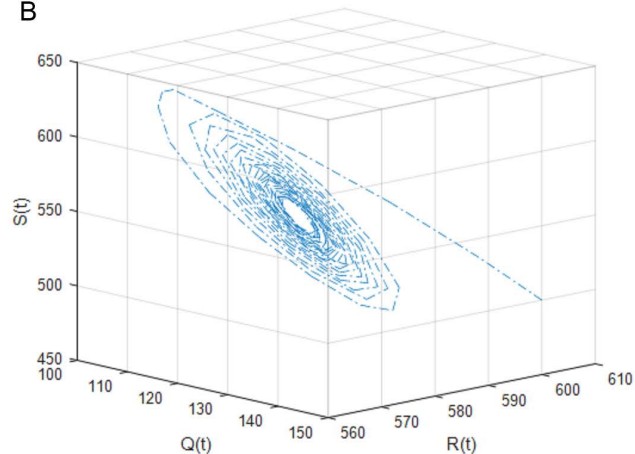

**Fig 5. When** $\tau_1 = 85 > \tau_{11} = 76, \tau_2 = 0$, **the positive equilibrium point** $P^*$ **becomes unstable.** (a) $S, E, I, Q, R$ with respect to time $t$ (b) SQR phase diagram.

### 5.3. Systems with optimal control

This section performs a numerical simulation using the fourth-order Runge-Kutta method and the forward and backward scanning method to assess the efficacy of the suggested optimal control strategy. An appropriate control strategy can successfully stop the spread of computer viruses, as shown by the experimental simulation. Let $b = 10$, $p = 0.025$, $d = 0.01$, $\theta = 0.0015$, $\beta = 0.0005$, $\sigma = 0.0014$, $\omega = 0.0155$, $\xi = 0.01$, $\varepsilon = 0.014$, $\gamma = 0.03$, $\eta = 0.01$, $\delta = 0.03$, $\alpha_1 = 0.01$, $\alpha_2 = 0.018$, $S(0) = 467/567$, $E(0) = 5/567$, $I(0) = 5/567$, $Q(0) = 5/567$, $R(0) = 85/567$, $\mu = 0 \sim 1$, $t_0 = 0$, $t_f = 1000$. The optimal control function diagram is obtained, as shown in Fig 10a. And Fig 10b shows the value of the target function $J$ at different $\mu(t)$. Fig 10c shows the changes of infected nodes over time under different control degrees. It can shows the change of infected nodes over time under different control levels. It can be seen from the Fig that when $\mu(t) = 0.0$, that is, without any control, the spread of computer viruses is uncontrollable. Although the number of infected nodes seems to be lower than the optimal control when $\mu(t) = 0.6$, $\mu(t) = 0.8$, more importantly, the optimal control strategy can minimize the value of the objective function $J$ and achieve the balance between control objective and control cost. Table 2 shows the values of the objective function $J$ of V. MadhuSudanan's SIRS model [28], Fangfang Yang's SIQRS model [29] and

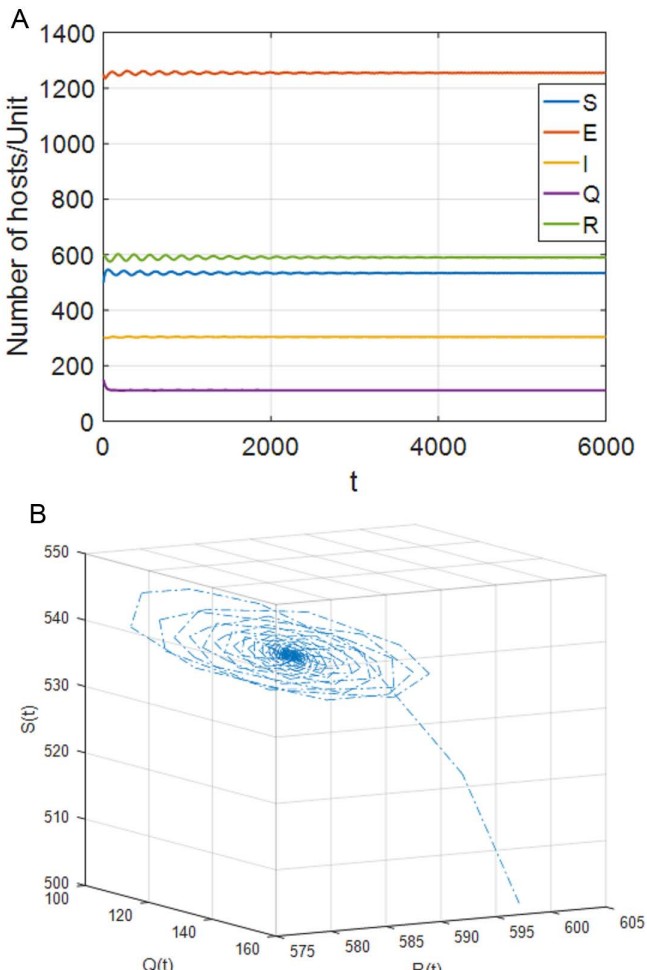

**Fig 6. When** $\tau_1 = 0, \tau_2 = 65 < \tau_{22} = 69$, **the positive equilibrium** $P^*$ **is locally stable.** (a) $S, E, I, Q, R$ with respect to time $t$ (b) SQR phase diagram.

SEIQRS model relative to the five constant control and optimal controls. Fig 10d reflects the change in the number of infected nodes in different models over time under optimal control. According to these data, we can see that the value of an objective function $J$ under optimal control is the smallest, and it can be seen that the time-varying optimal control strategy is more effective in controlling computer infection and virus spread. It can be seen from Table 2 and Fig 10d that after adopting the optimal control scheme, the three systems of SIRS [28], SIQRS [29] and SEIQRS can all achieve good control effect with minimal consumption.

At the same time, to verify the reliability of the proposed SEIQRS model and the effectiveness of optimal control, we conduct corresponding experiments on a real-time network. In the simulation experiments, we randomly set the initial positions and velocities of the nodes. We determine the initial infection rate by randomly selecting a subset of media and infecting them completely, while the remaining nodes are in a completely healthy state.

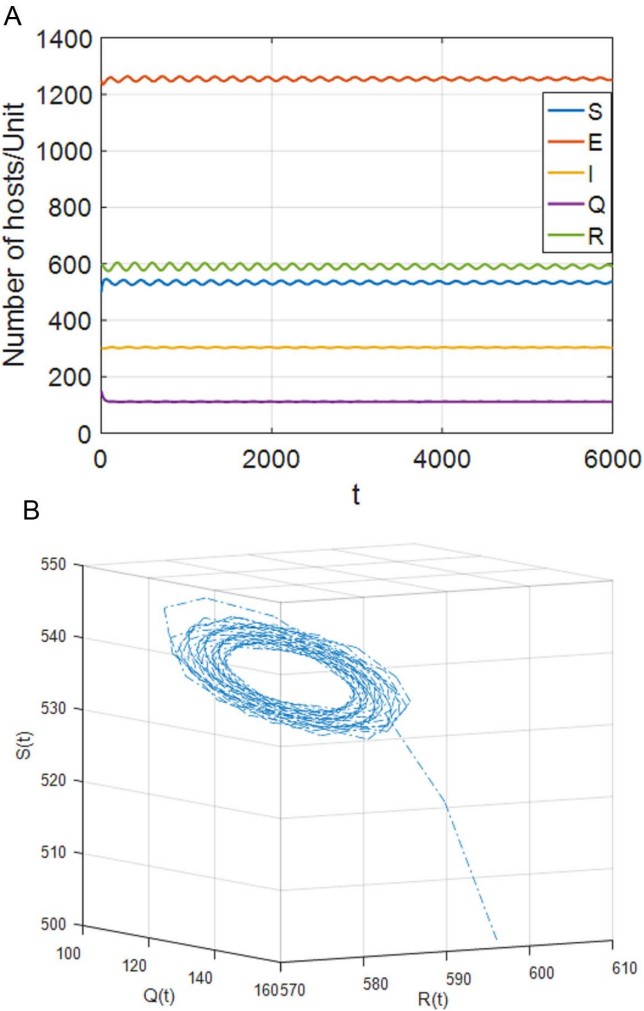

**Fig 7. When** $\tau_1 = 0, \tau_2 = 73 > \tau_{22} = 69$, **the positive equilibrium point** $P^*$ **becomes unstable.** (a) $S, E, I, Q, R$ with respect to time $t$ (b) SQR phase diagram.

In this section, we create a network system with 100 nodes, where each node is connected to any 4 nodes in the network graph. We randomly select five nodes from the system to be infected with the virus. Each node in the network structure is assigned a certain healing rate and infection rate, with the healing rate set to 1 and the infection rate set to 0.1. We set the maximum number of time steps to 1000, and within a certain number of time steps, all nodes in the system eventually get healed. This is shown in Fig 11. We use Matlab to conduct simulation experiments on the computer virus model, demonstrating the dynamic process of node infection in the computer virus model. Since the experimental results are a dynamic changing process, in this section, we capture four stages from the dynamic change graph to demonstrate the process of node infection and healing. Fig 11 shows the four stages of node infection and healing in the dynamic network system. (a) represents the initial stage, (b)(c) represent the intermediate stages, and (d) represents the final stage.

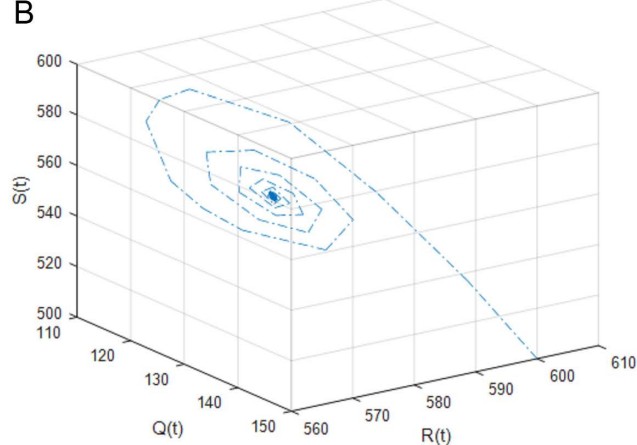

**Fig 8. When** $\tau_1 = \tau_2 < \tau_{**} = 30.6$, **the positive equilibrium** $P^*$ **is locally stable.** (a) $S, E, I, Q, R$ with respect to time $t$ (b) SQR phase diagram.

In Fig 11, we show the changes in susceptible and infected nodes for each stage. Susceptible nodes are shown in gray, latent nodes in yellow, infected nodes in red, isolated nodes in blue, and recovered nodes in green. In Fig 11, (a) shows the initial state of the simulation experiment. Four nodes are randomly selected from the network structure as already infected nodes, shown in red. The remaining uninfected nodes are shown in green. (b)(c) show the intermediate stages of the simulation experiment, illustrating the process of infection for uninfected nodes and the process of healing for already infected nodes. (d) shows the final stage of the simulation experiment, where the previously infected nodes have been healed and all nodes are shown in gray.

Through the simulation experiment with 100 nodes shown in Fig 11, we demonstrate the dynamic process of node infection and healing in the computer virus model, providing a theoretical visualization basis for computer virus control.

As the states of nodes in the dynamic network change, the number of infected and susceptible nodes in the network system also changes over time. Fig 12 shows the line graphs of the changing quantities of susceptible and infected nodes over time in the dynamic graph of the computer virus model simulation experiment. The green line represents the change

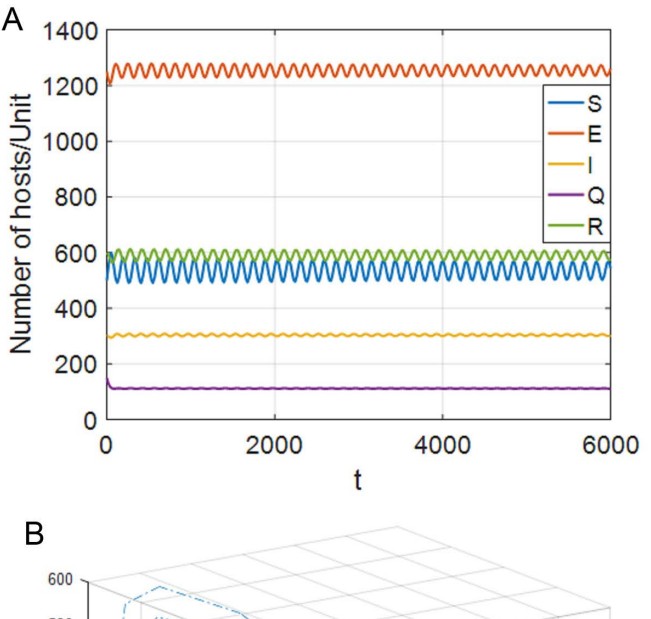

A

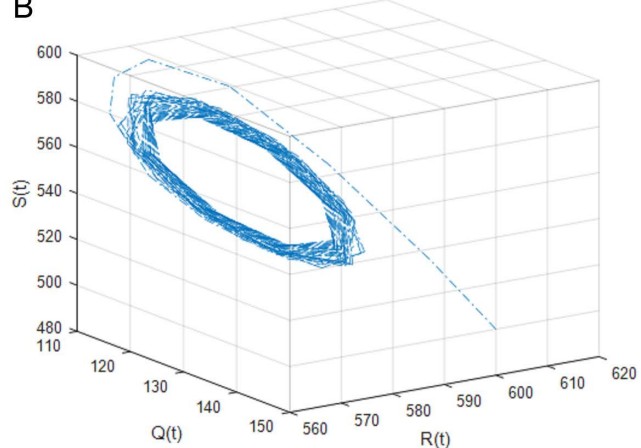

B

**Fig 9. When** $\tau_1 = \tau_2 > \tau_{**} = 30.6$, **the positive equilibrium point** $P^*$ **becomes unstable.** (a) $S, E, I, Q, R$ with respect to time $t$ (b) SQR phase diagram.

in the quantity of susceptible nodes, while the red line represents the change in the quantity of infected nodes. The changing curves in Fig 12 illustrate the process of the changing quantities of susceptible and infected nodes in the dynamic graph shown in Fig 11.

In particular, (a)(b)(c)(d) correspond to the node quantity change curves extracted from the four images in Fig 11. (a) represents the change in the quantity of infected and susceptible nodes in the initial state. (b)(c) represent the changing quantities of susceptible and infected nodes in the intermediate states of the system. (d) represents the changing quantities of susceptible and infected nodes in the final state of the system. From the changes in the red curve, it can be observed that under the initial conditions, when five nodes in the computer virus simulation system are infected and infect nearby nodes at a certain infection rate, the quantity of infected nodes increases while the quantity of susceptible nodes decreases. However, under certain healing rate conditions, when all the infected nodes are eventually healed, the system enters a stable state.

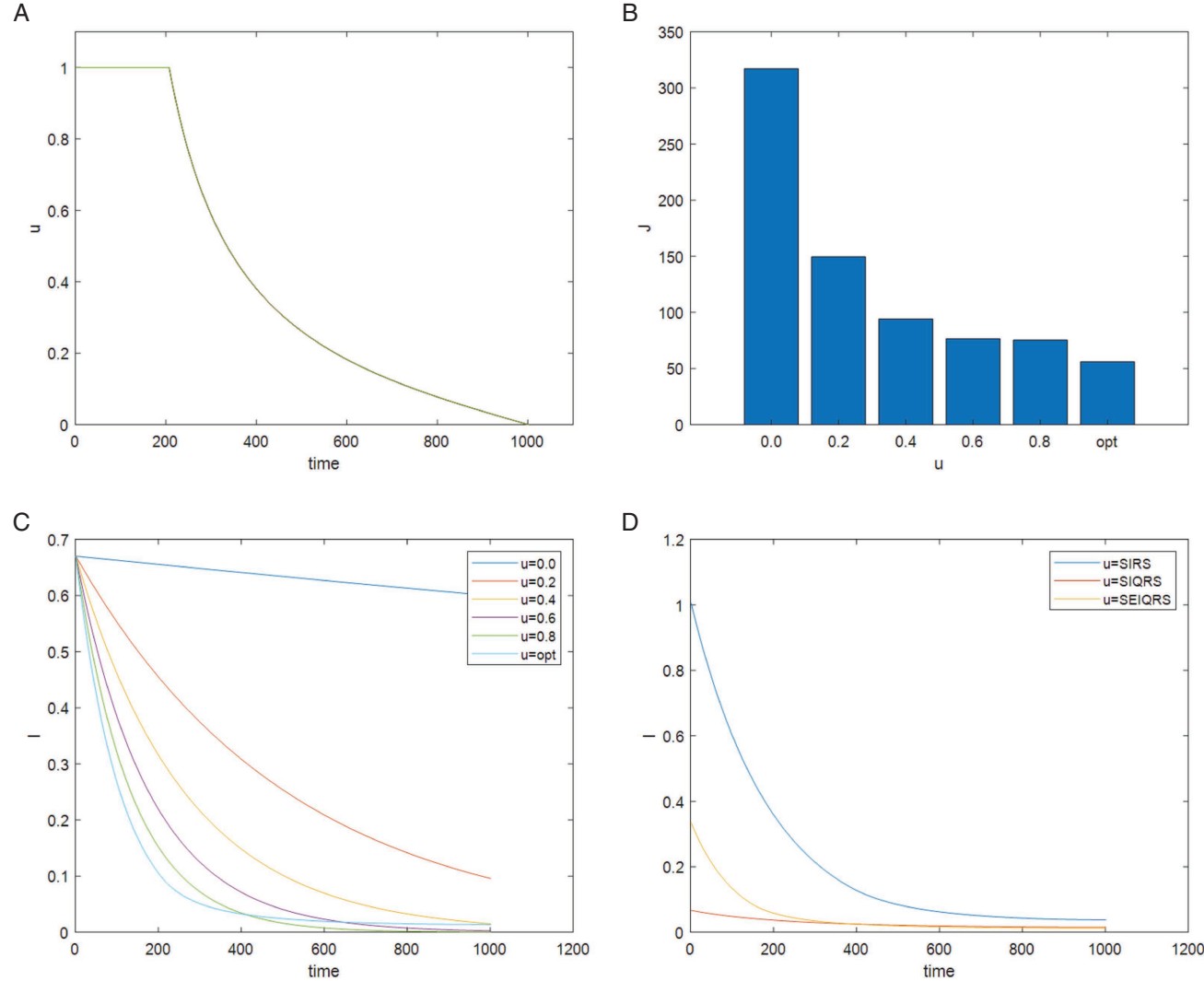

**Fig 10. Optimal control diagram.** (a) Control variable $\mu(t)$ as a function of time $t$ (b) The value of the target function $J$ at different $\mu(t)$ (c) Change of infected node $I$ with time $t$ under different $\mu(t)$ (d) Changes in the number of infected hosts of different models under optimal control.

**Table 2. The value of the objective function $J$ of different models under different control variable $\mu(t)$.**

| Variable of control $\mu(t)$ | 0.00 | 0.20 | 0.40 | 0.60 | 0.80 | opt |
|---|---|---|---|---|---|---|
| SIRS [28] $J$ | 467.50097 | 299.45854 | 212.89711 | 168.93011 | 148.98951 | 127.44763 |
| SIQRS [29] $J$ | 31.15207 | 21.83035 | 21.37881 | 27.28561 | 38.347903 | 18.539095 |
| SEIQRS $J$ | 165.18501 | 77.21131 | 51.45359 | 47.07925 | 53.011392 | 35.062202 |

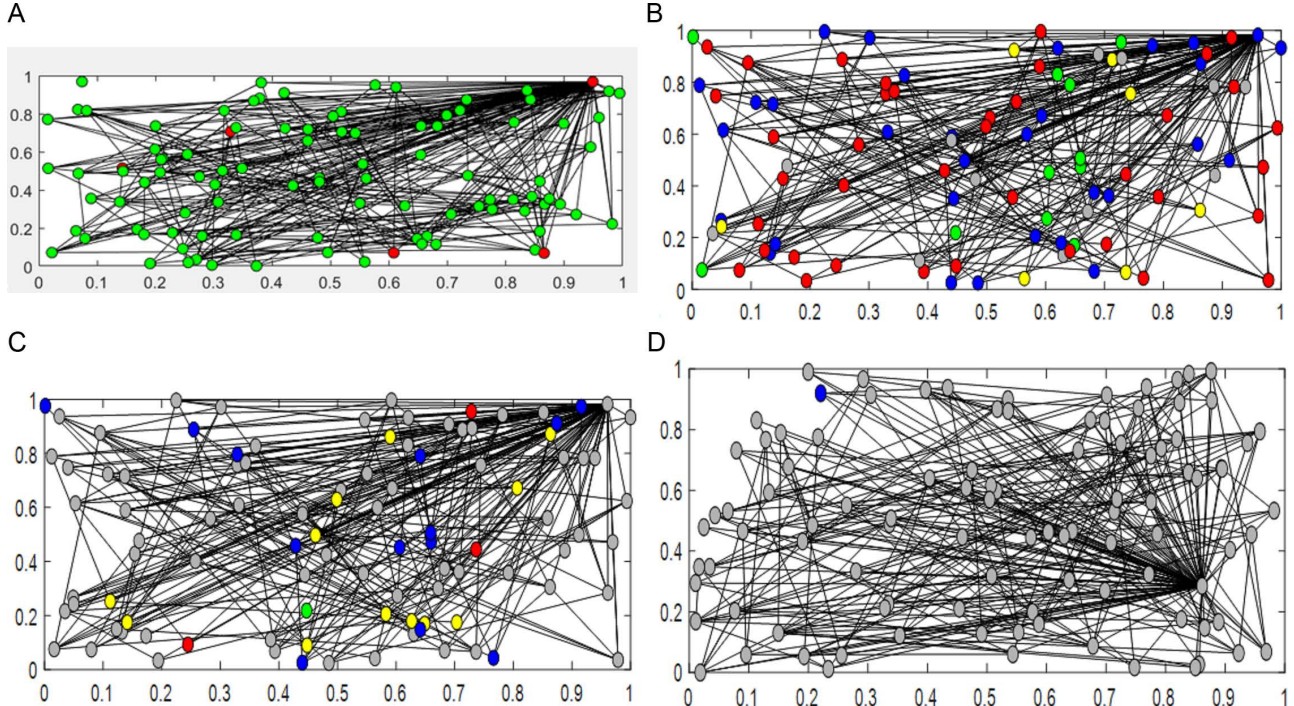

**Fig 11. Dynamic network node transformation diagram.** (a) The initial state (b) The intermediate stages (c) The intermediate stages (d) The final stage.

## 6. Conclusion

This paper presents a class of computer virus transmission models with nonlinear infection rates and double delay SEIQRS. The primary regeneration number $R_0$ is obtained through an in-depth analysis of the model, and the local stability of disease-free equilibrium is proved. The regional stability of positive balance under different time delays and Hopf bifurcation are discussed. And on this basis, the optimal control problem of this model is studied.

These results show that when the primary regeneration number $R_0$ is less than 1, the system will eventually stabilize at the disease-free equilibrium point. As time goes on, the infected node converges to 0, and the virus will eventually be eliminated. The system will eventually stabilize at the positive equilibrium point when the primary regeneration number is greater than 1 and the time delay is less than the critical value. At this time, the installation of anti-virus software and other measures can effectively inhibit the spread of computer viruses. Once the time delay exceeds the critical value, Hopf bifurcation appears in the system, and the spread of the virus will become uncontrollable. Under the optimal control condition, the trend of the number of infected computers is in line with the expectation of the objective function. The time-varying optimal control strategy can control the infection of computers and the spread of the virus more effectively. In the absence of control, the spread of computer viruses is uncontrollable. Meanwhile, applying different controls in SIRS [28], SIQRS [29], and SEIQRS proves that optimal control achieves the balance between control objectives and control costs. And at the same time, we carry out corresponding experiments on the real-time network to verify the reliability of the proposed SEIQRS model and the effectiveness of the optimal control.

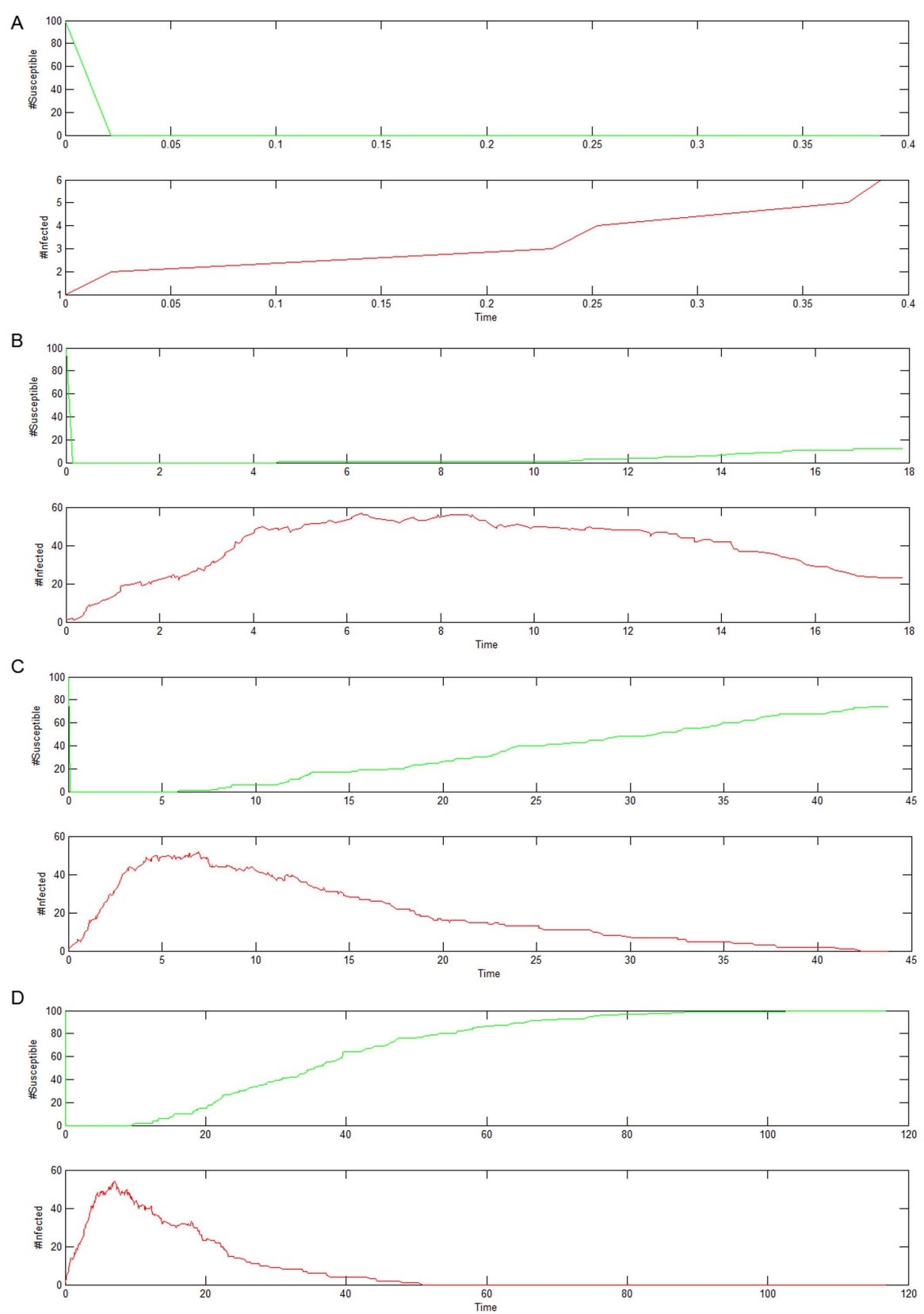

**Fig 12. Change curve of susceptible and infected nodes in the system.** (a) The initial state (b) The intermediate stages (c) The intermediate stages (d) The final stage.

## Author contributions

**Data curation:** Xinxin Chang.

**Visualization:** Xinxin Chang.

**Writing – original draft:** Lei Zhong.

**Writing – review & editing:** Junling Wang.

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
