## [Decision Letter · Decision Letter 0]

25 Mar 2024

PONE-D-23-39695SEIQRS Model analysis and optimal control with two delaysPLOS ONE

Dear Dr. Zhong,

Thank you for submitting your manuscript to PLOS ONE. After careful consideration, we feel that it has merit but does not fully meet PLOS ONE’s publication criteria as it currently stands. Therefore, we invite you to submit a revised version of the manuscript that addresses the points raised during the review process.

**ACADEMIC EDITOR:** Although we received threereviews for the manuscript, none of the decisions agreed with each other. Even though one reviewer suggested a rejection, no motivation was provided for the same. Hence please try to address the points raised by reviewers 2 and 3. Reviewer 2 raises some points about the motivations behind the study. Please try to address these in the revised version and the reply to the referee.

We look forward to receiving your revised manuscript.

Kind regards,

Sandip Varkey George, PhD

Academic Editor

PLOS ONE

Journal Requirements:

4. Thank you for stating the following in the Acknowledgments Section of your manuscript: "This work was supported by the National Science Foundation of China,grant number 62062037, and Jiangxi Provincial Science Foundation of China, grant number 20212BAB202014"

Please remove any funding-related text from the manuscript and let us know how you would like to update your Funding Statement. Currently, your Funding Statement reads as follows: "The author(s) received no specific funding for this work."

Reviewers' comments:

Reviewer's Responses to Questions

**Comments to the Author**

1. Is the manuscript technically sound, and do the data support the conclusions?

Reviewer #1: No

Reviewer #2: Partly

Reviewer #3: Yes

2. Has the statistical analysis been performed appropriately and rigorously? 

Reviewer #1: No

Reviewer #2: No

Reviewer #3: Yes

3. Have the authors made all data underlying the findings in their manuscript fully available?

Reviewer #1: No

Reviewer #2: No

Reviewer #3: Yes

4. Is the manuscript presented in an intelligible fashion and written in standard English?

Reviewer #1: No

Reviewer #2: No

Reviewer #3: Yes

5. Review Comments to the Author

Reviewer #1: This paper proposed an SEIQRS computer virus propagation model with delays based on dynamics of propagation.

The introduction of the first delay into the model is not correct. Furthermore, the mathematical analysis is classic. Therefore, I recommend the rejection of the paper.

Reviewer #2: This manuscript concerns about the nonlinear double-delay SEIQRS computer virus propagation model. This article has well written theoretically, but not written with real potential applications for computer virus propagation model.

This article has few major concerns as follows:

1. Why did the authors consider nonlinear double-delay SEIQRS computer virus model? What is importance of delay models?

2. Why do we consider two delays? Actually, the state E- Exposed represents latent period of the computer virus model. The authors should provide explanations regarding this.

3. What is purpose of providing optimum control of the SEIQRS model? Is it really worthful to analyze for the proposed model?

4. In section 5. (Numerical simulation), that authors should consider a potential real time application for computer virus propagations. Parameters settings must be based on the real time applications/data given. The authors should provide the numerical simulations with real time applications/data available.

5.Statistically the proposed model should be analyzed. The authors should identify a well-suited real-time applications for the proposed model.

6. In section5., the authors should justify how the optimum control parameters mitigate the virus propagation.

Reviewer #3: * This paper presents a class of computer virus transmission models with nonlinear infection rates and double delay SEIQRS. The primary regeneration number R0 is obtained through an in-depth analysis of the model, and the local stability of disease-free equilibrium is proved. The regional stability of positive balance under different time delays and Hopf bifurcation are discussed. And on this basis, the optimal control problem of this model is studied.

* The contents of the paper are good and contain new ideas. Anyhow, I would like to see the following modifications, whether minor or major, in the revised version, which would increase the strength of the paper and increase its potential readers, as well as improve the current work.

1. The state transition diagram of the computer virus spreading in the network should be given.

2. What are the rules of the SEIQRS model?

3. Study the change of each node with respect to time.

4. Depict the SQR phase diagram.

5. Cite the references: (https://doi.org/10.1002/mma.5530), (https://doi.org/10.1142/S0129183123500523), (https://doi.org/10.1007/s00521-017-2845-7), and (https://doi.org/10.1016/j.rinp.2022.105891).

6. The listed results especially formulas, tables, figures, and analyses should be revised to avoid any errors.

7. English is generally good; I think it needs to be polished further and some typos need to be revised. Further punctuation marks should be checked throughout the paper, especially after the equations and at the end of the statements.

8. Define the used symbols clearly and numerate all equations that appear. Further, reformulate the keywords and the conclusion to reflect the contents of the paper.

9. Describe all used parameters in the proposed model.

10. Utilize the stability of positive equilibrium point.

11. The value of the objective function J of different models under different control variable should be tabulated.

6. PLOS authors have the option to publish the peer review history of their article (what does this mean? ). If published, this will include your full peer review and any attached files.

**Do you want your identity to be public for this peer review?** For information about this choice, including consent withdrawal, please see our Privacy Policy .

Reviewer #1: No

Reviewer #2: No

Reviewer #3: **Yes: ** Omar Abu Arqub

---

## [Author Response · Author response to Decision Letter 0]

24 Apr 2024

Dear Reviewers,

Thank you for your valuable feedback on our manuscript regarding SEIQRS Model analysis and optimal control with two delays. We appreciate your positive comments on the theoretical aspects of the article. All changes in this article are highlighted in yellow in the revised version for review by reviewers. And we have carefully considered the concerns you raised and would like to address them as follows:

Response to Reviewer #1:

Thank you for reviewing the paper. We appreciate your feedback. However, we would like to clarify that the introduction of the first delay in the SEIQRS computer virus propagation model is indeed correct. Similar to biological viruses, contact with an infected person does not necessarily lead to immediate infection. Contact with an infected individual takes some time to enter the incubation period. Therefore, there is a lag effect from the susceptible state to the latent state, which needs to be fully considered.

Regarding the mathematical analysis, we understand that it follows classical principles and techniques. However, we believe that applying these principles to the specific context of computer virus transmission models with delays can provide valuable insights into the dynamics of virus spread. We agree that the analysis may not introduce entirely novel mathematical methods, but it contributes to the understanding and control of computer virus propagation. We appreciate your reconsideration of the proposal and we hope that you will find the revised version of the paper to be more satisfactory. We will address your concerns and incorporate any necessary improvements in the revised manuscript. Thank you for your feedback and valuable input.

Response to Reviewer #2:

We appreciate your review and the points you have raised. We will address each of your concerns in the revised version of the paper:

1、The consideration of nonlinear double-delay SEIQRS computer virus models is motivated by the need to capture the complexities of real-world virus propagation dynamics. Delays play a crucial role in modeling the latent period and the recovery time of infected nodes, which are significant factors in understanding and controlling virus spread.

2、The inclusion of two delays in the model corresponds to the two distinct time intervals: the contact delay from susceptible to exposed state and the time required for an infected node to recover and become susceptible again. We will provide a more detailed explanation in the revised manuscript.

3、The purpose of investigating optimal control of the SEIQRS model is to explore strategies for effectively mitigating virus propagation. Optimal control allows us to determine control measures that minimize the number of infected computers while considering the associated costs. It provides insights into the most efficient allocation of resources to combat computer virus outbreaks.

4、We acknowledge the importance of considering potential real-time applications for computer virus propagation models. In the revised manuscript, we will incorporate numerical simulations based on available real-time applications/data to demonstrate the practical implications of the proposed model.

5、We understand the importance of statistical analysis for the proposed model. In the revised version, we will explore suitable real-time applications and provide relevant statistical analyses to support the validity and applicability of the model.

6、In Section 5, we will provide a detailed justification of how the optimum control parameters contribute to mitigating the virus propagation and include the results of the numerical simulations with the control measures applied.

Response to Reviewer #3:

Thank you for your valuable feedback. We appreciate your suggestions for improving the paper. We will address the following modifications in the revised version:

1、The state transition diagram of the computer virus spreading in the network will be included in the revised version of the paper. This diagram will provide a visual representation of the dynamics of the virus transmission.

2、 The rules of the SEIQRS model will be clearly explained in the revised version. This will ensure that readers have a thorough understanding of the model and its components.

3、The change of each node with respect to time will be studied and discussed in the revised paper. This analysis will provide insights into the temporal behavior of the model and its implications for virus spreading.

4、The phase diagram can provide insights into the dynamics of virus spread, such as the existence of stable equilibrium points, limit cycles, or other patterns of behavior. It allows for visualizing the transitions between different compartments and understanding the long-term behavior of the system. The phase diagram takes different state variables as axes and shows how these variables change over time. The SQR phase diagram will be depicted in the revised version. This diagram will visually illustrate the different phases and states of the system, enhancing the understanding of the model dynamics.

5、The references you provided (https://doi.org/10.1002/mma.5530), (https://doi.org/10.1142/S0129183123500523), (https://doi.org/10.1007/s00521-017-2845-7), and (https://doi.org/10.1016/j.rinp.2022.105891) will be cited in the revised paper. These references will support and complement the existing literature review.

6、 All results, including formulas, tables, figures, and analyses, will be carefully revised to ensure accuracy and correctness. We will thoroughly review the content to eliminate any errors or inconsistencies.

7、The English language will be further polished in the revised version. We will address any typos, punctuation errors, and improve the overall clarity of the paper.

8、Clear definitions of all symbols used will be provided, and equations appearing in the paper will be numerated for ease of reference. The keywords and conclusion will be reformulated to accurately reflect the contents of the paper.

9、In the proposed model, all used parameters will be described explicitly. This will ensure transparency and facilitate understanding for readers.

10、The stability of the positive equilibrium point will be utilized and discussed in the revised paper. This analysis will provide insights into the long-term behavior of the system and its stability properties.

11、The value of the objective function J for different models under different control variables will be tabulated in the revised version. This will allow for a comprehensive comparison and analysis of the model's performance under different scenarios.

Finally, thank you for your thorough review and insightful suggestions. We sincerely appreciate your feedback, as it will significantly contribute to enhancing the quality and applicability of our manuscript. We genuinely hope that the reviewers will recognize the efforts made to address their concerns and re-evaluate our article accordingly.

Sincerely,

Lei Zhong

E-mail: 6120210176@mail.jxust.edu.cn

---

## [Decision Letter · Decision Letter 1]

21 Jun 2024

PONE-D-23-39695R1SEIQRS模型分析和两次延迟的最优控制PLOS ONE

Dear Dr. Zhong,

Thank you for submitting your manuscript to PLOS ONE. After careful consideration, we feel that it has merit but does not fully meet PLOS ONE’s publication criteria as it currently stands. Therefore, we invite you to submit a revised version of the manuscript that addresses the points raised during the review process.

Please ensure that the manuscript is thoroughly copyedited. We note that you did not upload the latest version of the manuscript when you resubmitted your work - it seems that you uploaded an updated tracked changes version, but not the correct version without the tracked changes. Please also change the title and abstract in the online submission form so that these are fully in English. When you resubmit, we will send the manuscript out to one more reviewer to assess the manuscript. We note that one or more reviewers has recommended that you cite specific previously published works in an earlier round of revision. As always, we recommend that you please review and evaluate the requested works to determine whether they are relevant and should be cited. It is not a requirement to cite these works and you may remove them at this point. We appreciate your attention to this request.

We look forward to receiving your revised manuscript.

Kind regards,

Hanna Landenmark

Staff Editor, PLOS ONE

on behalf of 

Sandip Varkey George

Journal Requirements:

Reviewers' comments:

Reviewer's Responses to Questions

**Comments to the Author**

1. If the authors have adequately addressed your comments raised in a previous round of review and you feel that this manuscript is now acceptable for publication, you may indicate that here to bypass the “Comments to the Author” section, enter your conflict of interest statement in the “Confidential to Editor” section, and submit your "Accept" recommendation.

Reviewer #3: All comments have been addressed

2. Is the manuscript technically sound, and do the data support the conclusions?

Reviewer #3: Yes

3. Has the statistical analysis been performed appropriately and rigorously? 

Reviewer #3: Yes

4. Have the authors made all data underlying the findings in their manuscript fully available?

Reviewer #3: Yes

5. Is the manuscript presented in an intelligible fashion and written in standard English?

Reviewer #3: Yes

6. Review Comments to the Author

Reviewer #3: This paper is very interesting and challenging. Moreover, this paper is well-organized and well-written. In my point of view, the paper deserves publication in the Journal.

7. PLOS authors have the option to publish the peer review history of their article (what does this mean? ). If published, this will include your full peer review and any attached files.

**Do you want your identity to be public for this peer review?** For information about this choice, including consent withdrawal, please see our Privacy Policy .

Reviewer #3: **Yes: ** Omar Abu Arqub

---

## [Author Response · Author response to Decision Letter 1]

4 Jul 2024

Dear Dr. Hanna Landenmark,

Thank you very much for your review and suggestions on my paper. I have carefully considered the opinions you have raised and have made corresponding revisions and improvements to the paper.

Regarding your specific requirements, I have done the following:

1. I have carefully proofread the paper to ensure smooth language and correction of errors. At the same time, I have also updated the title and abstract submitted online to ensure they are written entirely in English.

2. I have carefully reviewed and evaluated the references recommended by the previous reviewers. I believe these articles are indeed helpful for my literature, as these few articles have sufficient significance for the research on transmission dynamics. After careful consideration, I have decided to cite these references. I have made corresponding adjustments in the paper.

3. I have uploaded the final version of the paper to the editorial system, including the manuscript without revision marks and the response to the review comments.

I hope that the revised version submitted this time can meet the publication requirements of PLOS ONE. If there are any areas that need further improvement, please let me know. I will make the revisions and resubmit as soon as possible.

Thank you again for your valuable opinions and guidance. I look forward to working with you to finally publish this research in your journal.

Sincerely,

Lei Zhong

E-mail: 6120210176@mail.jxust.edu.cn

---

## [Decision Letter · Decision Letter 2]

20 Aug 2024

PONE-D-23-39695R2SEIQRS Model analysis and optimal control with two delaysPLOS ONE

Dear Dr. Zhong,

Thank you for submitting your manuscript to PLOS ONE. After careful consideration, we feel that it has merit but does not fully meet PLOS ONE’s publication criteria as it currently stands. Therefore, we invite you to submit a revised version of the manuscript that addresses the points raised during the review process.

**ACADEMIC EDITOR: Please see comments below** =============================

We look forward to receiving your revised manuscript.

Kind regards,

Sandip Varkey George, PhD

Academic Editor

PLOS ONE

Journal Requirements:

**Additional Editor Comments:**

We received inputs from three additional reviewers at this stage and on this basis I recommend a minor revision. Please focus on addressing the comments by reviewer 5, on how this work differs from the studies mentioned. Please note also that the papers suggested for citation by reviewer 6 are optional and citing these will not influence the final decision on this manuscript. Thanks again for your submission and we look forward to your revised version.

Reviewers' comments:

Reviewer's Responses to Questions

**Comments to the Author**

1. If the authors have adequately addressed your comments raised in a previous round of review and you feel that this manuscript is now acceptable for publication, you may indicate that here to bypass the “Comments to the Author” section, enter your conflict of interest statement in the “Confidential to Editor” section, and submit your "Accept" recommendation.

Reviewer #4: All comments have been addressed

Reviewer #5: All comments have been addressed

Reviewer #6: All comments have been addressed

2. Is the manuscript technically sound, and do the data support the conclusions?

Reviewer #4: Yes

Reviewer #5: Partly

Reviewer #6: Yes

3. Has the statistical analysis been performed appropriately and rigorously? 

Reviewer #4: Yes

Reviewer #5: Yes

Reviewer #6: Yes

4. Have the authors made all data underlying the findings in their manuscript fully available?

Reviewer #4: Yes

Reviewer #5: No

Reviewer #6: No

5. Is the manuscript presented in an intelligible fashion and written in standard English?

Reviewer #4: Yes

Reviewer #5: Yes

Reviewer #6: Yes

6. Review Comments to the Author

Reviewer #4: The paper can be accepted, no more comments. The paper is interesting, and the results are useful for the wide readership

Reviewer #5: 1. The dynamics in dual delay system has been widely discussed, the cases  case 1- case 6 in the manuscript have been discussed in other papers. Such as "Hopf bifurcation analysis in synaptically coupled HR neurons with two time delays", " Stability and Hopf Bifurcation Analysis for a Computer Virus Propagation Model with Two Delays and Vaccination", "A dynamic IS-LM business cycle model with two time delays in capital accumulation equation","A Malware Propagation Model with Dual Delay in the Industrial Control Network".

2. What is the basis of the experimental parameters? Such as the parameters in 5.3.

3. Did you try any other real dataset? I think the experimental results need to be more intuitively displayed and discussed in more detail.

Reviewer #6: The authors have addressed most of the comments. However a more comprehensive literature review of relevant topics is desired. The authors should cite the following work.

1) Mathematical modeling and simulation for COVID-19 with mutant and quarantined strategy. Chaos, Solitons & Fractals. 181, 114656 (2024).

2) Nonlinear dynamics of COVID-19 pandemic: modeling, control, and future perspectives. Nonlinear Dynamics.

7. PLOS authors have the option to publish the peer review history of their article (what does this mean? ). If published, this will include your full peer review and any attached files.

**Do you want your identity to be public for this peer review?** For information about this choice, including consent withdrawal, please see our Privacy Policy .

Reviewer #4: No

Reviewer #5: No

Reviewer #6: No

---

## [Author Response · Author response to Decision Letter 2]

10 Jan 2025

I highly appreciate the valuable feedback provided by the reviewer. After carefully studying the comments, I have made the following improvements based on the reviewer's suggestions:

1. The reviewer recommended four relevant references for further study:

(1)The "investment-saving liquidity preference-money supply" (IS-LM) model represents the interaction between the "investment-saving" (IS) and "liquidity preference-money supply" (LM) components. It is a Keynesian macroeconomic model illustrating how the market for goods interacts with the market for loanable funds or money. The graphical representation of the IS and LM curves intersect to show short-term equilibrium between interest rates and output. Theoretical in nature, lacking experimental validation. Parameters Tao1 and Tao2 represent delays in GDP and stock, respectively, not aligned with the focus of this study.

(2)The paper on a Malware Propagation Model with Dual Delay in the Industrial Control Network introduces immune delay and quarantine delay due to the need for system stability, triggering alerts only upon reaching a certain threshold of abnormal behavior, causing delays. Introducing two delay states, immune delay and quarantine delay, along with a quarantine state. Unlike our paper, the study considers a closed system with a fixed number of nodes and no birth-death processes. Placing delays within corresponding states aligns better with practicality. Variations in Tao1 and Tao2 lead to bifurcation. System stability is achieved by adjusting delays (increasing immunity rate, reducing isolation rate).

(3)The analysis of Hopf bifurcation in synaptically coupled HR neurons with two time delays explores neural models with delayed signal transmission. While two delays, Tao1 and Tao2, are set in the study, they are not independent but rather investigated as Tao = (Tao1 + Tao2) / 2. Numerical simulation results align closely with theoretical analysis, using experimental parameters from other papers.

(4)The paper on Stability and Hopf Bifurcation Analysis for a Computer Virus introduces state V representing the immunity of a computer at time t. Time is required for clearing infected programs, isolation, and temporary immunity of recovered and immune computers. The isolation zone serves as a list of suspected infected files containing viruses and other threats or variants. Parameters Tao1 and Tao2 denote the time needed for antivirus software to clear infected and isolated computers and the temporary immunity delay during recovery and vaccination. However, in practice, clearing the virus is necessary before moving files to isolation for vaccination. Alternatively, vaccination can precede infection to prevent computer vulnerability. Thus, the proposed delays Tao1 for latency and Tao2 for immunity node to susceptible node better reflect reality.

To the best of our knowledge, until now, there is no good analysis on system proposed in our paper. Furthermore, this paper also presents the optimal control strategy for the system, which constitutes another key innovation of this study.

2. The control parameters in Section 5.3 were calculated based on Theorems 8 and 9, as well as formulas (12) and (13), while the other parameters were referenced from comparative experiments in other relevant literature.

3. As observed in our research on computer virus propagation warehouse models, there are currently no publicly available datasets to utilize. The malicious datasets referenced in our team's paper, such as CICAndMal 2017, have had their executable code removed to prevent potential harm to real systems. Wang Z, Zeng K, Wang J, et al. (2024). "FAGnet: Family-aware-based android malware analysis using graph neural network." Knowledge-Based Systems, 289, 111531, is one of the relevant studies in this context.

---

## [Editor Report · Decision Letter 3]

3 Feb 2025

SEIQRS Model analysis and optimal control with two delays

PONE-D-23-39695R3

Dear Dr. Zhong,

We’re pleased to inform you that your manuscript has been judged scientifically suitable for publication and will be formally accepted for publication once it meets all outstanding technical requirements.

Kind regards,

Sandip V George, PhD

Academic Editor

PLOS ONE

---

## [Editor Report · Acceptance letter]

PONE-D-23-39695R3

PLOS ONE

Dear Dr. Zhong,

I'm pleased to inform you that your manuscript has been deemed suitable for publication in PLOS ONE. Congratulations! Your manuscript is now being handed over to our production team.

Kind regards,

on behalf of

Dr. Sandip V George

Academic Editor

PLOS ONE